

**Dynamics and environmental drivers of methane and nitrous oxide fluxes at the soil and ecosystem levels in a wet tropical forest**

**Authors**

Laëtitia M. Bréchet[1], Mercedes Ibáñez[1], Robert B. Jackson[2], Benoît Burban[1], Clément Stahl[1], Damien Bonal[3], Ivan A. Janssens[4]

[1]INRAE, UMR EcoFoG, CNRS, Cirad, AgroParisTech, Université des Antilles, Université de Guyane, Kourou, FR-97310, France

[2]Department of Earth System Science, Woods Institute for the Environment, and Precourt Institute for Energy, Stanford University, Stanford, CA 94305-2210, USA

[3]Université de Lorraine, AgroParisTech, INRAE, UMR Silva, Nancy, FR-54000, France

[4]Research Group Plant and Ecosystems (PLECO), Department of Biology, University of Antwerp, Wilrijk, BE-2610, Belgium

**Correspondence:** Laëtitia M. Bréchet (laeti.brechet@gmail.com)

**Abstract**

Tropical forests are critical for maintaining the global carbon balance and mitigating climate change, yet their exchange of greenhouse gases with the atmosphere remains understudied, particularly for methane ($CH_4$) and nitrous oxide ($N_2O$). This study reports on continuous measurements of $CH_4$ and $N_2O$ fluxes at the ecosystem and soil levels, respectively through eddy covariance and an automated



chamber technique, in a wet tropical forest in French Guiana over a period of 26 months. We studied
the magnitude of $CH_4$ and $N_2O$ fluxes and their drivers (climatic variables) during two extreme periods,
the driest and wettest seasons. Seasonal ecosystem fluxes showed near-zero net $CH_4$ uptake during
the driest season and emissions occurring during the wettest season that were larger in magnitude
than the uptake. Meanwhile, $N_2O$ emissions were of similar magnitudes in both seasons. Some upland
soils within the footprint of the eddy covariance tower emitted $N_2O$ in both seasons, although these
fluxes were particularly small. None of the measured climatic variables could explain this soil $N_2O$ flux
variation. In contrast, the upland soils were characterised by $CH_4$ uptake. Overall, seasonal ecosystem
$CH_4$ and $N_2O$ fluxes, as well as seasonal upland soil $CH_4$ fluxes, were partially explained by seasonal
variations in soil water content and global radiation. In addition to the upland soil fluxes studied, the
magnitude and sign of the net ecosystem fluxes of $CH_4$ and $N_2O$ were likely due to outgassing from
aboveground biomass and the presence of seasonally flooded areas within the footprint of the eddy
covariance system. Further studies of other ecosystem compartments in different forest habitats are
needed to better understand the temporal variations in $CH_4$ and $N_2O$ fluxes in wet tropical forests.



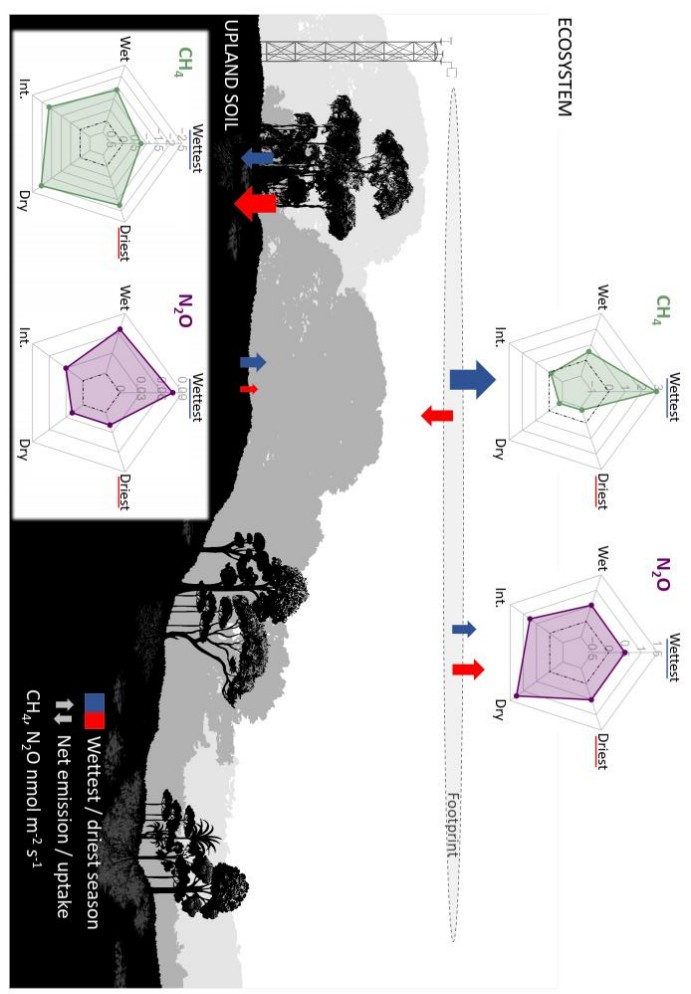

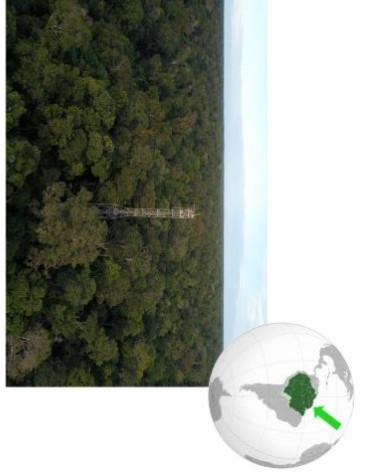

The 26-month study in a wet tropical forest revealed seasonal CH₄ and N₂O fluxes at the ecosystem and soil levels. Daily means of CH₄ and N₂O fluxes were highly variable, changing direction and magnitude on short time scales.



## 1 Introduction


The lack of knowledge on greenhouse gas fluxes in the tropical forests of the Amazon Basin contributes
significantly to the uncertainty in the global greenhouse gas budget, particularly for methane ($CH_4$) and
nitrous oxide ($N_2O$) (Davidson et al., 2012; Covey et al., 2021), the two most important greenhouse
gases in the atmosphere after carbon dioxide ($CO_2$). Early observations show that tropical forests in
the Amazon Basin may contribute disproportionately to global $CH_4$ and $N_2O$ exchanges compared to
other forests (Tian et al., 2015), but considerable uncertainties remain due to the paucity of data and
lack of detailed understanding of $CH_4$ and $N_2O$ cycling at both soil and ecosystem levels in these forests.
The role of tropical forest soils is crucial here as they can act either as a source or a sink for $CH_4$ and
$N_2O$ (Bouwman et al., 1993). In contrast to the consistent emissions from soil microbial decomposition
and root activity for $CO_2$, anaerobic $CH_4$-emitting microbes (methanogenic archaea) are dominant in
wetland environments, whereas aerobic $CH_4$-consuming microbes (methanotrophic bacteria) are more
abundant in upland soils (Ito and Inatomi, 2012; Welch et al., 2019) where they play the role of $CH_4$
sinks. For $N_2O$ as well, both emission and uptake can occur in soils. $N_2O$ can be produced by microbes
under both anaerobic (via denitrification) and aerobic (via nitrification) conditions (Khalil et al., 2004),
although the majority of $N_2O$ production occurs in waterlogged soils (Oertel et al., 2016). On the other
hand, soil microbes that are not denitrifiers can reduce $N_2O$ to dinitrogen (Sanford et al., 2012; Jones
et al., 2014).
Microbial $CH_4$ and $N_2O$ fluxes in tropical soils are controlled by the complex interplay of multiple
environmental and biological factors. The key factors regulating net $CH_4$ fluxes in tropical soils include
redox potential and water table depth (Silver et al., 1999; Teh et al., 2005; von Fischer and Hedin,
2007), plant productivity (Whiting and Chanton, 1993; von Fischer and Hedin, 2007), labile soil organic
matter (Wright et al., 2011), competition for carbon substrates among anaerobic microorganisms (Teh
and Silver, 2006; von Fischer and Hedin, 2007), temperature (Knox et al., 2021), and the presence of
plants that facilitate atmospheric escape (Pangala et al., 2013). The key factors regulating net soil $N_2O$
fluxes in tropical soils include redox potential, soil water content (SWC) or water table depth,



temperature, pH, labile carbon availability and labile nitrogen availability (Groffman et al., 2009). For
both $CH_4$ and $N_2O$ flux dynamics, of all these factors, variations in soil redox conditions, mediated by
variations in water table depth, play a particularly important regulatory role in tropical soils (Zhu et al.,
2013; Yu et al., 2021) due to the underlying physiology of the microbes that produce and uptake $CH_4$
and $N_2O$.
Production and uptake of both $CH_4$ and $N_2O$ in the soil are highly variable in space (hot spots) and time
(hot moments) (Blagodatsky and Smith, 2012). This is because microbial processes are discontinuous
(Blagodatsky and Smith, 2012), environmental conditions can change rapidly at short timescales, and
the strong seasonality of climate conditions, with pronounced wet and dry seasons, in most tropical
forests can significantly affect physical and ecophysiological ecosystem processes, which in turn affect
greenhouse gas fluxes. In addition, most published $N_2O$ and $CH_4$ flux data from tropical ecosystems
have been derived from chamber-based measurements at the soil level, often with low spatial and
temporal resolutions. Automated soil chambers capture fine-scale temporal variations, including hot
moments. However, they represent only a tiny part of the landscape (i.e. a few square metres of soil
surface at most) and therefore fail to capture emergent ecosystem properties that may be manifest at
larger spatial scales. Moreover, above-ground plant tissues also exchange $CH_4$ and $N_2O$ with the
atmosphere (i.e. produced in the soil and transported in the transpiration stream and/or by diffusion,
or produced within the stems), and this cannot be captured by soil chamber measurements alone. This
makes chamber approaches insufficient to quantify the magnitude and seasonal pattern of whole-
ecosystem greenhouse gas fluxes. Chamber-based measurements also hamper our ability to assess the
role of tropical forests in the exchange of $CH_4$ and $N_2O$ between the atmosphere and the land surface,
and induce large uncertainties in our current assessment of the greenhouse gas sink potential of
tropical forests.
On the other hand, a combination of soil and ecosystem level measurements can be a powerful tool
to reduce the gap between different levels of measurement (e.g. plot to ecosystem) (Lucas-Moffat et
al., 2018). Continuous ecosystem-level measurements via the eddy covariance technique provide high





temporal resolution data on mass and energy exchanges at the ecosystem level (Baldocchi, 2014, 2020;
Delwiche et al., 2021) and more detailed information on ecosystem functioning at a broader spatial
scale than do mere soil measurements, which miss above-ground exchanges and typically, also
emissions from wetland areas within the ecosystem (Bonal et al., 2008; Aguilos et al., 2018; Wang et
al., 2021; Liu et al., 2022). However, eddy covariance cannot indicate how much different land cover
types relatively contribute to the ecosystem's total flux since the measurements integrate high and
low frequency flows over time and space. Chamber and eddy covariance-based approaches each have
their own strengths and weaknesses; however, taken together, they effectively represent the
magnitude of ecosystem fluxes and can help determine the drivers of greenhouse gas flux dynamics
(Eugster et al., 2015). We therefore combined these two approaches to test the following assumptions:
• H1: Ecosystem- and soil-level $CH_4$ and $N_2O$ fluxes vary seasonally in the studied tropical forest,
switching between uptake and emission,
• H2: At both the soil and ecosystem levels, SWC is the primary abiotic driver of these gaseous
fluxes during the driest and wettest seasons.
This study provides, for the first time, a comprehensive assessment of $CH_4$ and $N_2O$ dynamics at both
ecosystem and soil levels based on high-frequency eddy covariance and continuous soil chamber time
series over 26 months in a wet tropical forest.

**2 Methods**
**2.1 Study site**
Our research was conducted at the Guyaflux site (5°16'54"N, 52°54'44"W) (Bonal et al., 2008), an ICOS-
associated ecosystem station (GF-Guy) located 15 km from the coast and approximately 40 km west of
Kourou, in French Guiana, South America. On a decadal time scale, the average annual precipitation at
the study site is 3102 ± 70 mm and average annual air temperature is 25.7 ± 0.1 °C (Aguilos et al., 2018).
The climate is humid tropical and highly seasonal due to the north-south movement of the Inter-
Tropical Convergence Zone (ITCZ), which drives regional precipitation. The ITCZ dictates the wet season



(from December to July, with rainfall of up to 500 mm month$^{-1}$) and the long dry season from mid-
August (mid-November, with less than 100 mm month$^{-1}$). In the northernmost part of the Guiana
shield, where the study site is located, the topography results in a succession of small elliptical hills
from 10 to 40 m asl, with soils classified as nutrient-poor acrisols (IUSS Working Group WRB, 2015).
The site is totally surrounded by undisturbed forest, locally characterised by a tree density of about
620 trees ha$^{-1}$ (for trees > 10 cm dbh), an average tree height of 35 m, an average tree diameter at
breast height (DBH) of 40.1 cm, with emergent trees over 40 m tall, and a tree species richness of about
140 species ha$^{-1}$ (Bonal et al., 2008; Aguilos et al., 2018; Daniel et al., 2023).

**2.2 Tower-based flux measurements**
Continuous measurements of the surface-atmosphere exchange of $CO_2$, $H_2O$ and energy were initiated
in 2003 based on the Euroflux methodology (Aubinet et al., 2000) and the eddy covariance approach
(Baldocchi, 2003); they have previously been reported and fully documented (Bonal et al., 2008;
Aguilos et al., 2018). The Guyaflux flux tower is 55 m high and extends about 20 m beyond the top of
the canopy. The putative average footprint of the eddy fluxes from the tower covers approximately 50
- 100 ha of undisturbed forest in the direction of the prevailing winds (Bonal et al., 2008; Fang et al.,
2024). Within the estimated footprint of the Guyaflux tower, 52% of the area is upland forest, 13% is
seasonally flooded forest and the rest (35%) is slope forest (Fig. S1). Most of the meteorological and
eddy flux sensors are mounted three meters above the top of the tower, and include equipment
measuring air temperature and humidity (HMP155, Vaisala, Helsinki, Finland), bulk precipitation
(ARG100, EM lmt, Sunderland, UK), wind direction and speed (A05103-5, Young, Traverse City, MI,
USA), and global infrared incident and reflected radiation (Rg) (CNR1, Kipp and Zonen, Bohemia, NY,
USA). All the meteorological data in the present study were collected at 1-min intervals and compiled
as 30-min averages or sums with data loggers (CR23X, CR1000 or CR3000 models; Campbell Scientific
Inc., Utah, USA).



In 2017, a closed-path fast greenhouse gas analyser (FGGA, Los Gatos Research, Mountain View,
California, USA), whose head (gas inlet) was mounted 0.3 m from the head of a 3-D sonic anemometer
(R3-50; Gill Instruments, Lymington, UK), was set up at the top of the eddy flux tower to provide eddy
covariance measurements of the $CH_4$ and $N_2O$ fluxes. The FGGA, equipped with a fourth-generation
cavity-enhanced laser absorption spectroscopy analyser (DLT-100; Los Gatos Research Inc.), was
connected to an external pump (Edwards XDS-35i, Edwards, England, UK) and to a 62 m long PFA inlet
tube (4 mm inlet diameter) protected by black foam with a 15 μm filter. All data were sampled at a
frequency of 20 Hz with data loggers (model CR3000; Campbell Scientific Inc.).
In addition, to take conditions where non-turbulent processes prevail (e.g. calm nights) into account,
the eddy covariance measurements were complemented with a vertical profile measurement system
to estimate variations in $CH_4$ and $N_2O$ concentrations at six different heights (i.e. 0.5, 6, 13, 23, 32 and
58 m) with a 0.8 L min$^{-1}$ pump connected to a six-line solenoid valve and a closed-path FGGA (FGGA,
Los Gatos Research, Mountain View, California, USA). The entire system was controlled by a data logger
(model CR10X; Campbell Scientific Inc.), which recorded greenhouse gas concentration data every 15
min. The vertical profile system for $CH_4$ and $N_2O$ was stopped after one year because the storage of
the gases was found to be negligible (see below).

**2.3 Tower-based $CH_4$ and $N_2O$ flux computation**

We used EDDYPRO V6.2.2 (LI-COR Inc.), a software based on a set of standardised post-processing
calculations and corrections, to calculate $CH_4$ and $N_2O$ fluxes from the raw high-frequency eddy
covariance data. The parameterization of the software included: a two-dimensional coordinate
rotation to set lateral and vertical mean wind speed to zero; a time lag between each scalar and wind
speed measurement estimated by covariance maximisation; an empirical frequency correction for
high-frequency attenuation; and a Webb-Pearman-Leuning correction for density fluctuations where
required, i.e. where concentrations were not measured as mixing ratios. Details of these corrections
are given in Aubinet et al. (2012). After the greenhouse gas flux computation, the EDDYPRO output



files contained continuous time series for ecosystem-atmosphere greenhouse gas ($CH_4$ and $N_2O$) fluxes
reported at a 30-min time step (from 17 May, 2016 to 2 August, 2018). The output files also included
uncertainties, quality control flags, friction velocity, and basic environmental and meteorological data.
To calculate net ecosystem production and uptake, we added the storage term to the turbulent flux
measured by the eddy covariance tower. This correction is particularly relevant for $CO_2$ exchanges in
forest ecosystems to reduce the uncertainty of the net flux estimate (Nicolini et al., 2018). However,
for the net $CH_4$ and $N_2O$ fluxes, the relevance of the storage term correction was only marginal. In
contrast to $CO_2$, whose concentrations clearly built up at soil level during low-turbulence conditions,
this was not the case for $N_2O$ and $CH_4$, and comparisons between the ecosystem fluxes with and
without correction for the storage term showed that the change in the resulting flux was minimal (Figs.
S2, S3). Consequently, we assumed that the storage of $CH_4$ and $N_2O$ was negligible and ignored it in
this study. This meant that a larger period of eddy covariance flux measurements could be used
(starting in 2016), in addition to the January 2017 - January 2018 period where $CH_4$ and $N_2O$ storage
data were available. Ecosystem fluxes of $CH_4$ and $N_2O$ were calculated every half-hour (nmol $m^{-2}$ $s^{-1}$).

**2.4 Chamber measurements**
In addition to the flux tower and its associated instrumentation, automated static non-steady through-
flow chambers for continuous measurement of soil greenhouse gas fluxes were installed in June 2016
on hypoferralic soils with deep vertical drainage and a very deep water table (~15 m depth),
approximately 50 m upwind from the flux tower in some of the upland forest part of the tower
footprint (Fig. S1). This automated system had two constraints, which when combined, limited the
spatial coverage of the soil greenhouse gas flux measurements to the upland forest area: the power
supply was only available at the flux tower, and the maximum distance between the automated
chambers and the gas analysers was 30 m. Thirteen of the sixteen initial chambers functioned correctly
throughout the study period and their data were retained in this study. Briefly, the chambers (LI-8100-
104, LI-COR Inc., Lincoln, NE, USA) were mounted on PVC collars (20.3-cm inner diameter; enclosed



soil area ~318 cm$^2$; offset ~4 cm) that were permanently inserted into the soil. The chambers were
connected to a multiplexer (LI-8150, LI-COR Inc.), used to program specific measurement cycles, which
operated with a cavity ring-down spectroscopy (CRDS) analyser (G2308; Picarro Inc., Santa Clara, CA,
USA) to measure $CO_2$, $H_2O$ and dry air-$CH_4$ and $N_2O$ concentrations (water corrected concentrations)
at 1 Hz. This analyser relied on an external recirculation pump (A0702; Picarro Inc.). The multiplexer
program purged the system 15 s before and 45 s after the measurements to flush out the tubing and
return to ambient-air greenhouse gas concentrations. A dead band of 60 s avoided potential
measurement errors ascribed to pressure changes inside the chamber-tubing-analyser loop following
chamber or solenoid valve closure and accounted for time lags. In addition, the program included two
different closure times to account for high and low fluxes, i.e. 2-min and 25-min measurement periods.
The equipment is described in more detail in previous publications (Courtois et al., 2019; Bréchet et
al., 2021).

**2.5 Chamber-based $CH_4$ and $N_2O$ flux computation**
We used the SOILFLUXPRO software (LI-COR Biosciences) to compute soil greenhouse gas fluxes based
on the linear and exponential regression of the change in headspace concentrations over time, the
collar area and the system volume, after correction for atmospheric pressure and temperature. Flux
values were selected based on the model that provided the best fit and highest determination
coefficient ($R^2$).
After calculating the fluxes and implementing our standard soil greenhouse gas QC procedure (Courtois
et al., 2019; Bréchet et al., 2021), all $CO_2$ fluxes with an insufficiently high $R^2$ (< 0.90), an initial
concentration greater than 900 ppm, or a value outside the range of variation from 0.10 to 30 μmol
m$^{-2}$ s$^{-1}$ were discarded for all three gases, based on the assumption that poor-quality $CO_2$ implied poor-
quality values for $CH_4$ and $N_2O$. As an improvement over Courtois et al. (2019), all $CH_4$ fluxes with $R^2$ <
0.80 were excluded regardless of the measurement length (i.e. 2-min and 25-min). For $N_2O$, all short
measurements (i.e. 2-min) with $R^2$ < 0.80 were discarded. In addition, based on the metric proposed



by Nickerson (2016), we calculated minimum detectable fluxes suitable for high-resolution in situ
greenhouse gas measurements as 0.040 nmol m$^{-2}$ s$^{-1}$ and 0.002 nmol m$^{-2}$ s$^{-1}$ for 2 min and 25 min
respectively for CH$_4$ and 0.100 nmol m$^{-2}$ s$^{-1}$ and 0.002 nmol m$^{-2}$ s$^{-1}$ for 2 min and 25 min respectively for
N$_2$O. The soil fluxes of CH$_4$ and N$_2$O (nmol m$^{-2}$ s$^{-1}$) were then assigned to the respective half-hours.

**2.6 Tower and chamber flux data analysis**
In order to include the most complete information possible, we based the study period on the soil flux
measurements and included all available data from 17 May, 2016 to 2 August, 2018. This 26-month
period included both very dry and very wet seasons (Fig. 1).



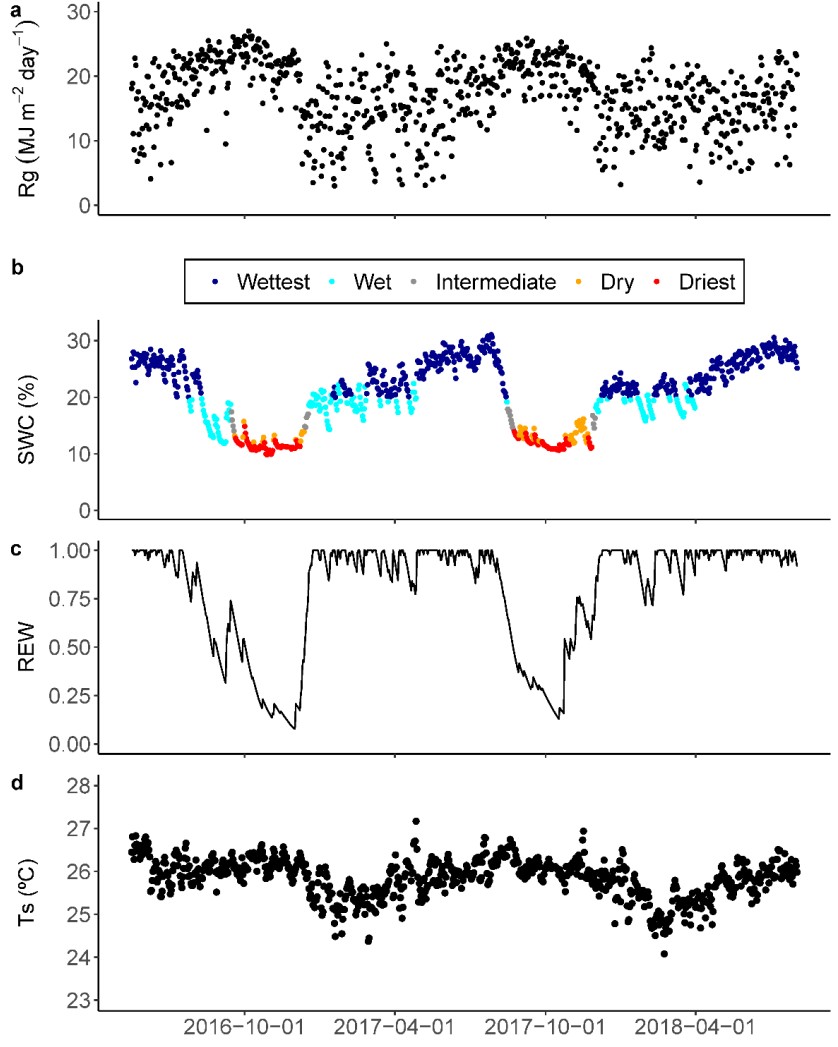


Figure 1. Daily (a) accumulated global radiation (Rg); (b) average soil water content (SWC) at 5 cm in

depth during the wet, intermediate and dry seasons, and for two contrasted seasons defined as the

wettest (dark blue dots) and the driest (red dots); (c) average relative extractable water (REW) to 3 m

in depth based on the water balance model developed by Wagner et al. (2011); and (d) soil

temperature (Ts) at 5 cm in depth, from 17 May, 2016 to 2 August, 2018 in the Guyaflux tropical forest,

French Guiana. See Sect. 2.7 for details of the methods used to define the "driest" and "wettest"

periods with extreme SWC.



Some flawed data was found (and eliminated) for both eddy covariance and soil chamber
measurements. They resulted from particular physical or biological conditions at the sampling point or
inside the soil chamber (e.g. wasp nests, disturbance by birds, dust, a branch preventing proper closure
of the chamber and causing a leak), or from mechanical issues (e.g. a power cut, soil chamber
remaining closed, gas analyser malfunction), which generated gaps in each time series. After flux
computation, the eddy covariance data for $CH_4$ and $N_2O$ were filtered: data below a u* threshold of
0.15 m s$^{-1}$ were discarded (Bonal et al., 2008) - as were data with a quality flag of 2 (on a scale from 0
to 2) (Mauder and Foken, 2004).
For eddy covariance and chamber data, the 30 min observations were filtered and flux values outside
the 5$^{th}$ - 95$^{th}$ percentile flux range were discarded. To calculate daily averages for greenhouse gas
fluxes, we first estimated the optimal number of observations per day necessary to obtain
representative daily averages. To do this, we selected a data pool with at least 42 observations per day
in the eddy covariance dataset. In the soil chamber dataset, we calculated daily means for each of the
thirteen chambers and retained only the data when at least five observations per chamber per day
were recorded. Subsets of values from 1 to 42 for the eddy covariance data and from 1 to 13 for the
soil chamber data were then created for each day based on 100 bootstrap iterations. Representative
daily means were found for thresholds of 12 minimum observations per day for eddy covariance and
10 for chamber data. These tests were performed separately for $CH_4$ and $N_2O$ and the driest and
wettest seasons, giving similar threshold results. Daily means with a number of observations below the
corresponding threshold were then discarded from further analyses. After filtering out the non-
representative days, the missing daily means for the whole study period represented 27% for both $CH_4$
and $N_2O$ flux data derived from eddy covariance, and 34% and 30%, respectively, for $CH_4$ and $N_2O$ flux
data derived from the soil chambers.

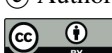



**2.7 Environmental measurements**
In the vicinity of the tower, we used temperature sensors (CS107; Campbell Scientific Inc., Logan, UT,
USA) to measure surface soil temperature (Ts) and frequency domain sensors (CS615 or CS616;
Campbell Scientific Inc.) to measure soil volumetric water content (SWC) at a depth of 5 cm. To
estimate the daily relative extractable water (REW) for trees from the soil surface to a depth of 3 m,
we used a soil water balance model previously validated for tropical forests (Wagner et al., 2011), with
daily precipitation, evapotranspiration and solar radiation as input variables. Daily SWC (%), Ts (°C) and
REW were defined as the average of the half-hourly flux values over 24 h, while dialy Rg (MJ m$^{-2}$ day$^{-1}$)
was the sum of the half-hourly flux values over 24 h.
To examine the effect of environmental variables on $CH_4$ and $N_2O$ fluxes at the ecosystem and soil
levels, we extracted data from two contrasting periods, termed "Driest" and "Wettest" (Fig. 1). The
driest days occurred at the end of the dry season, when SWC was less than 15% and decreased for at
least three consecutive days. The wettest days had a SWC above 20%, corresponding to a REW above
0.4, and unlimited available water for trees (Wagner el al., 2011) for more than two consecutive days.

**2.8 Data analysis**
We used the mgcv (Wood and Wood, 2015) and stats packages in R version V3.6.3 (R Core Team, 2020)
for our data analyses and ggplot2 for visualisations (Wickham and Wickham, 2016). The significance
level for all tests was set at 0.05.
We used Kolmogorov-Smirnov tests (ks.test function) to evaluate the effects of contrasting seasons,
specifically the driest and wettest periods, on the distributions of $CH_4$ and $N_2O$ fluxes at both ecosystem
and soil levels. A Student's t-test (t.test function) was used to determine if the greenhouse gas fluxes
were statistically different from 0. Generalised additive models (GAM; gam function) were used to
assess whether climate variables (i.e. Rg, Ts, SWC) explained the temporal variations in $CH_4$ and $N_2O$
fluxes at the ecosystem and soil levels. We included the default thin-plate spline smoothing parameter
selected by restricted maximum likelihood (REML), and modelled the fluxes of each greenhouse gas as



a function of season, climate variables and their interaction. For all GAMs, the "select" option was set
to TRUE so that terms could be removed from the GAM during model fitting if they provided no benefit
(Wood, 2017).

**3   Results**
**3.1 Environmental seasonality**
The Guyaflux site is characterised by an alternating wet and dry season, typical of a wet tropical
climate. During the wet season, mean daily global radiation (Rg; Fig. 1a) was at its lowest, while soil
water content (SWC; Fig. 1b) was at its highest, accompanied by peak values for relative extractable
water (REW; Fig. 1c). In contrast, the dry season had elevated mean daily Rg, minimal SWC and the
lowest values of REW. The soil temperature (Ts; Fig. 1d) also exhibited a clear seasonal pattern, albeit
weak in absolute values (approximately 2°C), which was influenced by changes in air temperature.
During the study period, the driest season (SWC ranging from 9.9% to 15.0%) covered 15.8% of the
total study period (128 days), while the wettest season (SWC ranging from 20.0% to 30.0%) covered
55% of the total study period (444 days) and represented near-saturated conditions.

**3.2 Greenhouse gas flux seasonality under contrasting environmental conditions**
The ecosystem and soil $CH_4$ and $N_2O$ fluxes also displayed some seasonality (Figs. 2, S4, S5), with
seasonal differences particularly evident between the wettest and the driest season (Fig. 3).



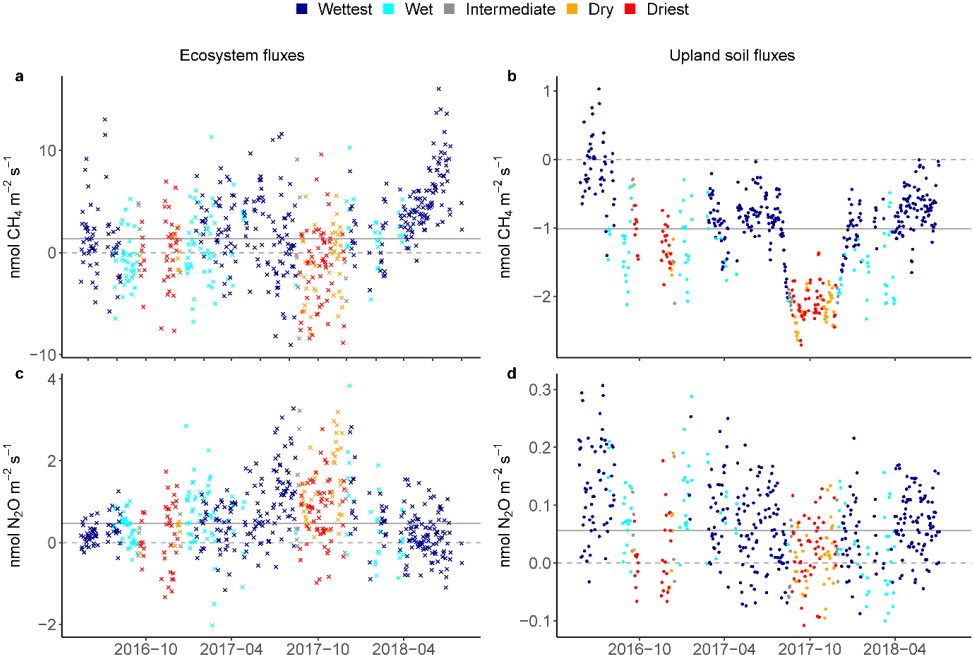

Figure 2. Seasonal courses of average daily ecosystem (crosses on the left) and upland soil (solid dots on the right) fluxes for the wet, intermediate and dry seasons, and for two contrasted seasons defined as the wettest (dark blue dots) and the driest (red dots) for 24-hour $CH_4$ fluxes (a, b) and $N_2O$ fluxes from 17 May, 2016 to 2 August, 2018 (c, d) in the Guyaflux tropical forest, French Guiana. Positive fluxes (above the dashed grey "0" line) indicate greenhouse gas emissions and negative fluxes (below the "0" line) indicate greenhouse gas uptake; the solid grey line represents the median over the whole period. Note that the scale of the y-axis has been adjusted for each gas and compartment to improve clarity.



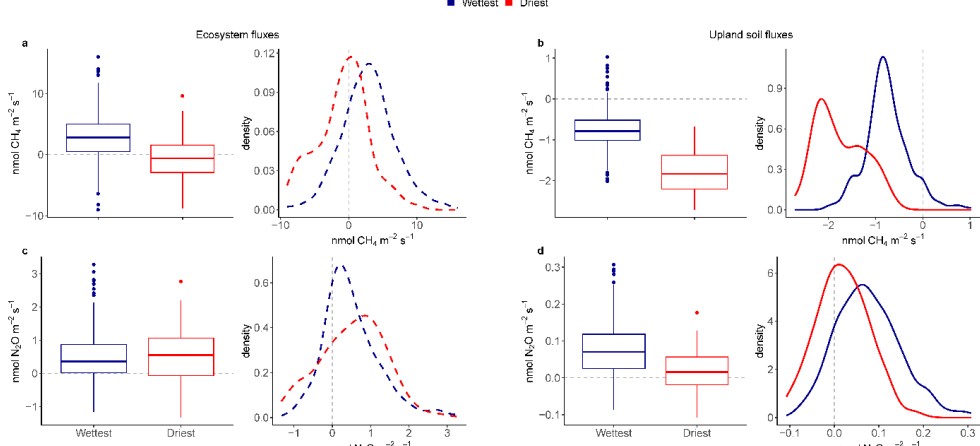

Figure 3. Boxplots and associated density plots of average daily ecosystem fluxes (dashed lines on the left) and upland soil (solid lines on the right) fluxes of 24-hour $CH_4$ fluxes (a, b) and $N_2O$ fluxes (c, d) for the wettest (blue) and driest (red) seasons, from 17 May, 2016 to 2 August, 2018 in the Guyaflux tropical forest, French Guiana. In the box plots, solid bold lines represent medians, box boundaries mark the 25th and 75th percentiles and whiskers show the 10th and 90th percentiles. Dots mark outliers. In the density plots, positive fluxes on the right side of the dotted "0" line indicate greenhouse gas emissions and negative fluxes on the left side of the "0" line indicate greenhouse gas uptake. All differences among fluxes in the wettest and driest season were statistically significant at $p < 0.05$. See Table 1 for the Kolmogorov-Smirnov test results.



$CH_4$ emissions were greater during the wettest season than during the driest season, when net fluxes
hovered around zero (Table 1; Fig. 3a). In contrast to the ecosystem-level fluxes, soil $CH_4$ fluxes in some
of the upland forest were mainly negative, indicating net soil $CH_4$ uptake throughout the year (Fig. 2b),
even under varying environmental conditions (Table 1; Fig. 3b). Soil $CH_4$ uptake did decreased
significantly in the wettest season compared to the driest season, although the fluxes remained
negative overall (i.e. $CH_4$ uptake, Table 1; Fig. 3b).

Table 1. Mean, standard deviation (SD) and median ecosystem and upland soil $CH_4$ and $N_2O$ fluxes for
the wettest and driest seasons in the Guyaflux tropical forest, French Guiana. Values in bold are
different from 0 at p level < 0.05 based on Student's t-test.

| Fluxes | Wettest | | | Driest | | |
|---|---|---|---|---|---|---|
| | Mean | SD | Median | Mean | SD | Median |
| Ecosystem flux (nmol$_{CH4/N2O}$ m$^{-2}$ s$^{-1}$) | | | | | | |
| **$CH_4$** | 2.9 | 3.9 | 2.8 | -0.8 | 3.8 | -0.6 |
| **$N_2O$** | 0.5 | 0.7 | 0.4 | 0.5 | 0.8 | 0.6 |
| Upland soil flux (nmol$_{CH4/N2O}$ m$^{-2}$ s$^{-1}$) | | | | | | |
| **$CH_4$** | -0.8 | 0.5 | -0.8 | **-1.8** | 0.5 | -1.8 |
| **$N_2O$** | 0.1 | 0.1 | 0.1 | 0.0 | 0.1 | 0.0 |


The seasonal pattern of ecosystem $N_2O$ fluxes was less pronounced than for $CH_4$ (Fig. 2c); the driest
season showed only slightly higher emissions than the wettest season (Table 1; Fig. 3c). In contrast to
the ecosystem-level fluxes, soil $N_2O$ fluxes in upland areas not only had a more pronounced seasonal
pattern, the upland soils also emitted more $N_2O$ during the wettest season than during the driest
season, when the average flux was near-zero $N_2O$ (Table 1; Fig. 3d). It is noteworthy that, although
there were significant differences between seasons for all fluxes and at both ecosystem and soil levels
(Fig. 3), the overall mean flux was significantly different from zero only for soil $CH_4$ fluxes during the
driest season (Table 1). This indicates that the magnitude of the fluxes was very low relative to the
large variability among the seasons.




**3.3 Environmental drivers of ecosystem and upland soil greenhouse gas fluxes**

Net ecosystem $CH_4$ fluxes showed that $CH_4$ emissions decreased with increasing Rg (Fig. 4a), although

this negative correlation was statistically significant only during the wettest season (Table 2) when net

emissions occasionally switched to net uptakes at highest Rg values (Fig. 4a). Net ecosystem $CH_4$ fluxes

were strongly positively correlated with SWC (Fig. 4b), showing increased $CH_4$ emissions with

increasing SWC, although this correlation too was statistically significant only during the wettest

season (Table 2). A weaker, yet statistically significant, correlation was detected between ecosystem

$CH_4$ fluxes and Ts (Table 2).



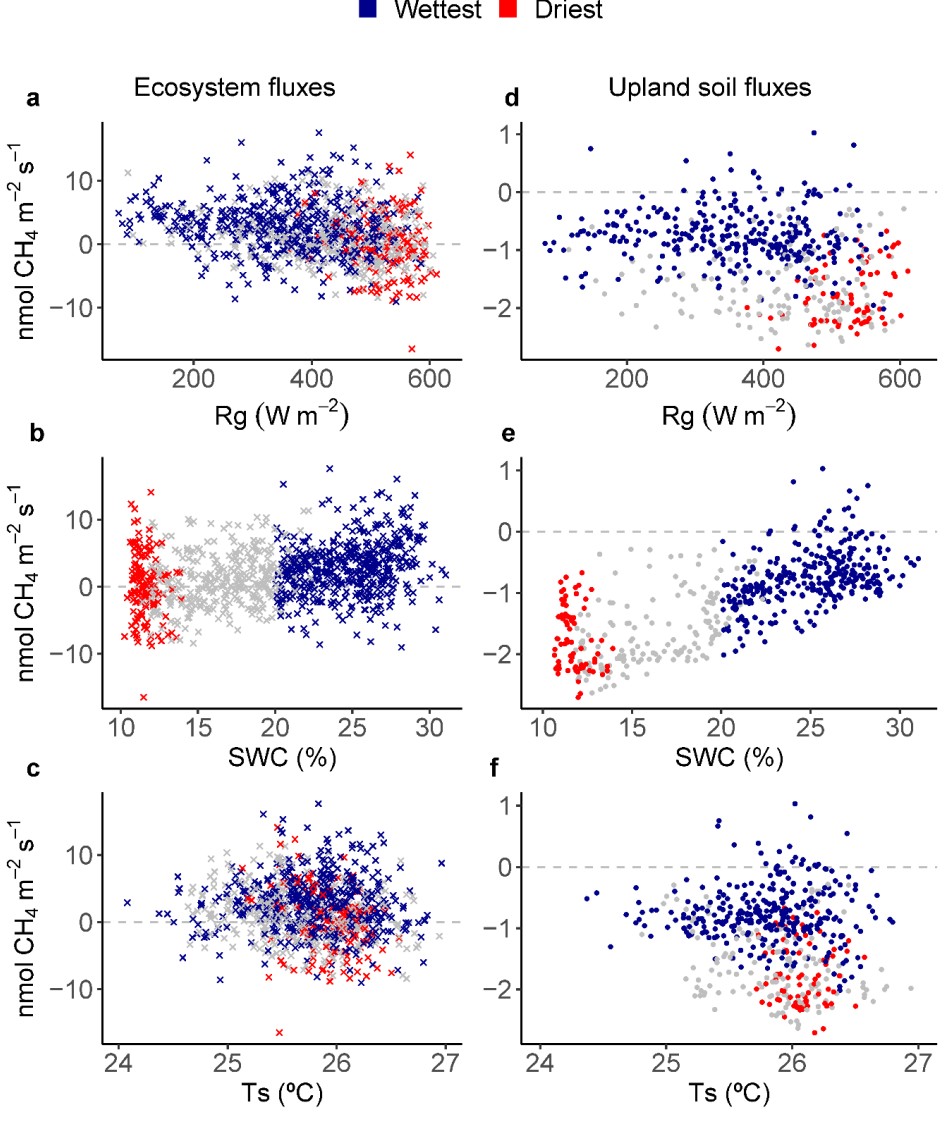

Figure 4. Relationships between environmental drivers (global radiation (Rg), soil water content (SWC)
and soil temperature (Ts)) and daily average ecosystem (crosses on the left) and upland soil (solid dots
on the right) $CH_4$ fluxes for the wettest (blue) and driest (red) seasons, with remaining data in grey,
from 17 May, 2016 to 2 August, 2018 in the Guyaflux tropical forest, French Guiana. Positive fluxes
above the horizontal "0" line indicate $CH_4$ emissions and negative fluxes below the horizontal "0" line
indicate $CH_4$ uptake.



Despite the different signs for the net $CH_4$ flux at ecosystem- and soil levels, relationships comparable
with the environmental drivers observed for ecosystem $CH_4$ fluxes were also found for some of the
upland soil: net soil $CH_4$ uptake increased with increasing Rg (Fig. 4d) and decreased with increasing
SWC (Fig. 4e). The correlation between upland soil $CH_4$ fluxes and Rg was statistically significant in the
driest season, while the correlation with SWC was equal and significant in both seasons (wettest and
driest, Table 2).
Ecosystem $N_2O$ fluxes showed relatively weak responses to the environmental drivers we investigated
(Figs. 5a-c). The statistically significant terms in the model were SWC > Rg >Ts, and only during the
wettest season. However, the $R^2$ of the model was rather low ($R^2$ = 0.04; Table 2). For the upland soil
$N_2O$ fluxes, none of the environmental drivers we investigated explained soil $N_2O$ emissions (Fig. 5;
Table 2).



Table 2. Results of generalised additive models (GAM) assessing the relationships between
environmental variables, i.e. global radiation (Rg), soil water content (SWC), soil temperature (Ts), and
daily mean ecosystem and upland soil $CH_4$ and $N_2O$ fluxes during the wettest and driest seasons from
17 May, 2016 to 2 August, 2018 in the Guyaflux tropical forest, French Guiana. The effective degrees
of freedom (edf) and the reference number of degrees of freedom (Ref. df) of the fitted models, with
values for each spline term, are shown. Significant terms at p level < 0.05 are shown in bold.

| | Fluxes | Best model predictors | $R^2$ | Intercept | Coefficients edf | Ref. df | F value | p value |
|---|---|---|---|---|---|---|---|---|
| **Ecosystem level** | | | | | | | | |
| Daily | $CH_4$ | | 0.20 | 0.002 | | | | |
| | | Rg: Wettest | | | 0.8 | 9 | 0.54 | **0.014** |
| | | Rg: Driest | | | 0.6 | 8 | 0.19 | 0.113 |
| | | Ts: Wettest | | | 1.7 | 9 | 0.62 | **0.025** |
| | | Ts: Driest | | | 0.0 | 9 | 0.00 | 0.477 |
| | | SWC: Wettest | | | 1.6 | 9 | 1.92 | **< 0.001** |
| | | SWC: Driest | | | 0.0 | 6 | 0.00 | 0.433 |
| Daily | $N_2O$ | | 0.04 | 0.000 | | | | |
| | | Rg: Wettest | | | 0.8 | 9 | 0.41 | **0.026** |
| | | Rg: Driest | | | 0.0 | 8 | 0.00 | 0.772 |
| | | Ts: Wettest | | | 1.2 | 9 | 0.39 | **0.045** |
| | | Ts: Driest | | | 0.3 | 8 | 0.05 | 0.247 |
| | | SWC: Wettest | | | 2.2 | 9 | 0.93 | **0.011** |
| | | SWC: Driest | | | 0.0 | 6 | 0.00 | 0.717 |
| **Upland soil level** | | | | | | | | |
| Daily | $CH_4$ | | 0.54 | -0.001 | | | | |
| | | Rg: Wettest | | | 1.2 | 9 | 0.24 | 0.156 |
| | | Rg: Driest | | | 1.5 | 9 | 1.24 | **0.001** |
| | | Ts: Wettest | | | 0.7 | 9 | 0.27 | 0.057 |
| | | Ts: Driest | | | 0.0 | 9 | 0.00 | 0.865 |
| | | SWC: Wettest | | | 2.6 | 9 | 9.74 | **< 0.001** |
| | | SWC: Driest | | | 1.9 | 7 | 3.52 | **< 0.001** |
| Daily | $N_2O$ | | 0.10 | 0.000 | | | | |
| | | Rg: Wettest | | | 0.0 | 9 | 0.00 | 0.419 |
| | | Rg: Driest | | | 0.6 | 8 | 0.19 | 0.112 |
| | | Ts: Wettest | | | 1.2 | 9 | 0.32 | 0.084 |
| | | Ts: Driest | | | 0.0 | 9 | 0.00 | 1.000 |
| | | SWC: Wettest | | | 0.0 | 9 | 0.00 | 0.804 |
| | | SWC: Driest | | | 0.0 | 6 | 0.00 | 1.000 |


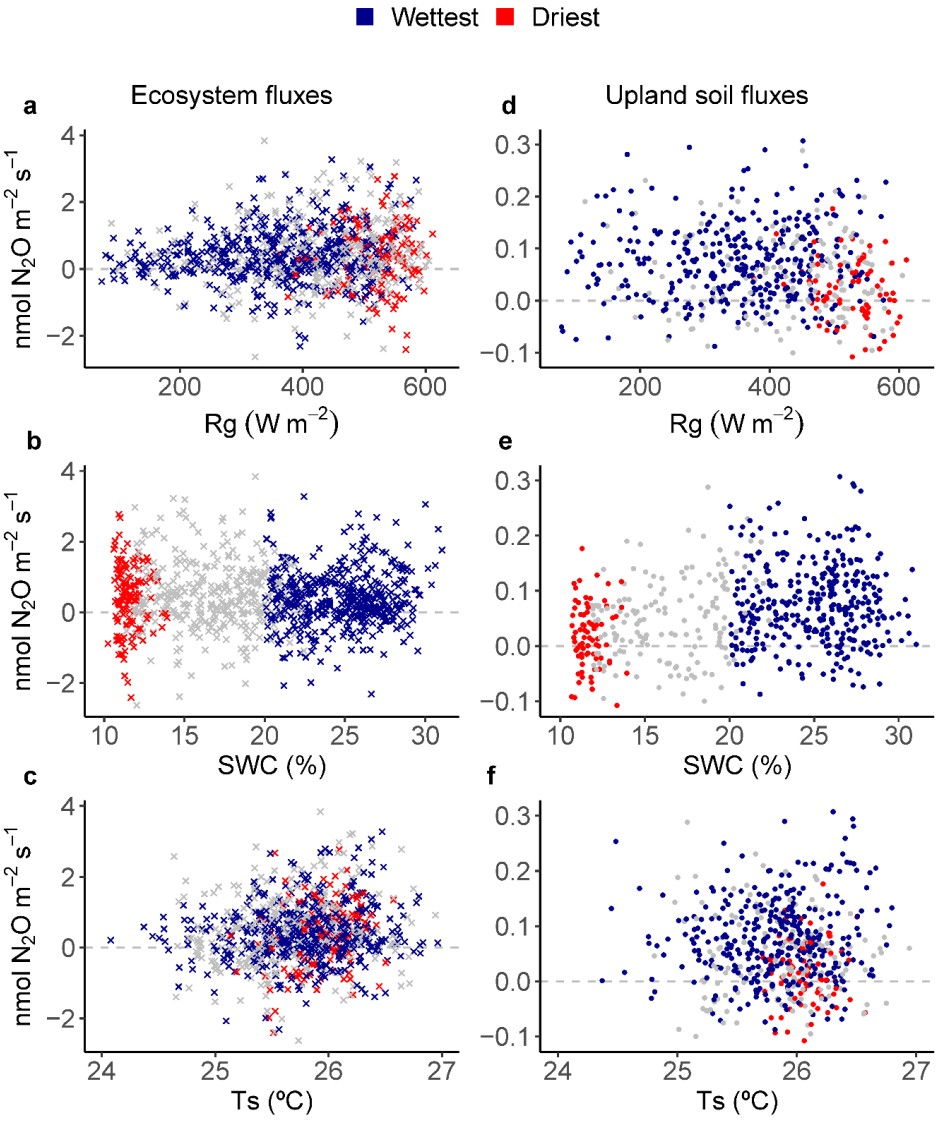

Figure 5. Relationships between environmental drivers (global radiation (Rg), soil water content (SWC) and soil temperature (Ts)) and daily average ecosystem (crosses on the left) and upland soil (solid dots on the right) $N_2O$ fluxes for the wettest (blue) and driest (red) seasons, with remaining data in grey, from 17 May, 2016 to 2 August, 2018 in the Guyaflux tropical forest, French Guiana. Positive fluxes above the horizontal "0" line indicate $N_2O$ emissions and negative fluxes below the horizontal "0" line indicate $N_2O$ uptake.



## 4   Discussion

To our knowledge, this is the first study to report on simultaneous ecosystem and upland soil $CH_4$ and

$N_2O$ flux observations in a wet tropical forest over a period of more than two years (Fig. 2). This study

provides a unique opportunity to investigate the dynamics and environmental drivers of $CH_4$ and $N_2O$

fluxes in these ecosystems.

### 4.1 Ecosystem and upland soil-$CH_4$ fluxes

4.1.1 Seasonal variations in ecosystem $CH_4$ fluxes: trends and drivers

Our long-term monitoring of eddy covariance $CH_4$ fluxes above the Guyaflux forest canopy showed

high temporal variability, with changes in the sign (net emission or uptake) and amount of the

ecosystem fluxes observed over short time scales, supporting hypothesis H1 (Fig. 2). Net $CH_4$ emission

rates ($2.9 \pm 3.9$ nmol$CH_4$ m$^{-2}$ s$^{-1}$; Mean $\pm$ SD) dominated during the wettest season, whereas net $CH_4$

uptake ($-0.8 \pm 3.8$ nmol$CH_4$ m$^{-2}$ s$^{-1}$) was more common during the driest season, although large

temporal variations occurred throughout the study seasons (Figs. 2-3, Table S1). Much higher wet-

season net fluxes had previously been found in two Brazilian tropical forests, Manaus and Sinop (62.3

and 34.6 nmol$CH_4$ m$^{-2}$ s$^{-1}$, respectively; Carmo et al., 2006), though the studies were based on canopy

air samples and a modelling approach. Surprisingly, these Brazilian forests acted as an even larger $CH_4$

source during the driest season (64.1 and 88.3 nmol$CH_4$ m$^{-2}$ s$^{-1}$, respectively; Carmo et al., 2006), while

the Guyaflux forest switched from a $CH_4$ source during the wet periods to a small sink during the dry

ones. Ecosystem $CH_4$ fluxes are driven by a combination of plant, microbial and abiotic processes,

which are mediated by both living and dead plants, and can explain episodic bursts (Eugster and Plüss,

2010; Covey and Megonigal, 2019). The mechanism underlying the large $CH_4$ emissions during the dry

season observed in the Brazilian forests remains unknown, but the authors suggest that it may have

been connected to the anaerobic decay of waterlogged wood, undrained soil patches or the

waterlogged cavities of tank bromeliads. Concomitantly, drought-induced reduced oxidation in the soil



surface layer may have exacerbated the net $CH_4$ emissions. Contrary to Carmo et al. (2006), Sakabe et
al. (2018) found a seasonal pattern similar to the one we observed in our study where the eddy
covariance technique was applied. Although the flux values they found had a higher range of variation
(10.3 $nmolCH_4$ $m^{-2}$ $s^{-1}$ versus -8.5 $nmolCH_4$ $m^{-2}$ $s^{-1}$, respectively, in the wet and dry seasons), this was
most likely due to the different ecosystem they studied, an Indonesian tropical peat swamp forest.
Consistent with H2, the generalised additive models (GAM) revealed that SWC, and to a lesser extent
Rg and Ts, were relevant ecosystem $CH_4$ flux drivers, particularly during the wettest season (Table 2;
Figs. 4a-c). It is reasonable to assume that high SWC but relatively low Ts during the wettest season
stimulated $CH_4$ production in most compartments of the ecosystem, not only in the seasonally flooded
soils. An increase in SWC at shallow depths may reduce the amount of air-filled pore space in the soil.
This reduction may decrease the diffusion of oxygen and $CH_4$ from the atmosphere through the soil to
methanotrophs, resulting in a decrease in net uptake or an increase in net emissions, if production
exceeds uptake (Wang et al., 2013). Such processes may occur in all the soil types within the footprint
of the eddy flux tower (see Sect. 4.1.2), and they may partially explain the seasonal trends observed at
our site. On the other hand, the statistically significant, albeit weak, relationship between $CH_4$
emissions and Rg during the wettest season could occur if the occasional high light intensity (Fig. 1a) is
sufficient to stimulate plant-mediated $CH_4$ transport through sap flow, and / or if the measured forest
area has more seasonally flooded areas than upland forest. The latter explanation is, however, unlikely
because the location of the Guyaflux tower (~300 m from the seasonally flooded area) was specifically
chosen to guarantee consistent types of ecosystem flux observations regardless of the season and of
associated changes in wind direction and atmospheric stability. Further research is needed to clarify
the correlation between Rg and net $CH_4$ flux. Increased fluxes in the flooded areas and anaerobic
microsites, rather than seasonal changes in the footprint, probably explain part of the observed
seasonal variations.
Disentangling the drivers of net $CH_4$ fluxes is further complicated by aboveground processes that also
contribute to $CH_4$ emissions and uptake in forest ecosystems. Soil-produced $CH_4$ dissolved in water can





indeed be taken up by roots, transported through the xylem stream in the stem, branches and leaves,
and then released into the atmosphere, thus bypassing the oxidation processes in the shallow soil
layers. As such, the highest $CH_4$ emissions from trees have been found in waterlogged soils, for
example, in wetland and riparian forests (Pangala et al., 2013; Covey and Megonigal, 2019; Gauci et
al., 2025). However, recent studies have shown that tree compartments (i.e. stems, branches, and
leaves) can also consume $CH_4$, particularly in free-draining upland soils (Gauci et al., 2024). At our study
site, both stem $CH_4$ emission and uptake were observed within the footprint of the Guyaflux tower
(Bréchet et al., 2021, 2025; Daniel et al., 2023). Although these fluxes were weak, they contributed to
the seasonal variations in ecosystem $CH_4$ exchanges (Bréchet et al., 2021, 2025; Daniel et al., 2023).

4.1.2 Seasonal variations in upland soil $CH_4$ fluxes: trends and drivers
The upland soils studied within the tower footprint were active consumers of atmospheric $CH_4$ (Fig.
3b), with overall net uptake rates of $1.1 \pm 0.7$ nmol$CH_4$ m$^{-2}$ s$^{-1}$, which is higher than the global average
for tropical forests ($-0.7$ nmol$CH_4$ m$^{-2}$ s$^{-1}$ or $-2.5$ kg$CH_4$-C ha$^{-1}$ yr$^{-1}$, Dutaur and Verchot, 2007) but lower
than fluxes found in previous studies in tropical plantations and forests in Central and South America
(Panama, $-13.5$ nmol$CH_4$ m$^{-2}$ s$^{-1}$, Keller et al., 1990; $-19.9$ nmol$CH_4$ m$^{-2}$ s$^{-1}$, Goreau and de Mello, 1988).
$CH_4$ fluxes at our site ranged seasonally from $-0.8 \pm 0.5$ nmol$CH_4$ m$^{-2}$ s$^{-1}$ in the wettest season to $-1.8 \pm$
$0.5$ nmol$CH_4$ m$^{-2}$ s$^{-1}$ in the driest season (Table S2), supporting H1 and globally corroborating other
seasonal studies in tropical forests. In addition, $CH_4$ flux dynamics in our upland soils were
characterised by a large range of variation, but a consistent sign, between the driest and wettest
seasons. A study conducted in a seasonal tropical forest in China with static chambers showed a
comparable seasonal pattern for soils: they acted mainly as $CH_4$ consumers, with an uptake rate of 0.7
$\pm 0.0$ nmol$CH_4$ m$^{-2}$ s$^{-1}$ (or $29.5 \pm 0.3$ µg$CH_4$-C m$^{-2}$ h$^{-1}$; Werner et al., 2006) during the dry period. The
uptake decreased by approximately 50% after the first rainfall events and the associated increases in
SWC. Another study carried out with the static chamber technique near the Guyaflux forest and in
similar environmental conditions reported that upland soils consumed $1.0 \pm 3.2$ nmol$CH_4$ m$^{-2}$ s$^{-1}$ during



the dry season (Courtois et al., 2018). Yet, those soils become slight emitters during the wet season
(0.1 ± 0.9 nmolCH$_4$ m$^{-2}$ s$^{-1}$; corresponding to -44.0 ± 139.7 µgCH$_4$-C m$^{-2}$ h$^{-1}$ and 3.7 ± 40.1 µgCH$_4$-C m$^{-2}$
h$^{-1}$ for the dry and wet seasons, respectively; Courtois et al., 2018). However, although meaningful,
these comparisons between studies should be interpreted with great caution because the
measurement techniques differed (i.e. automated in our study versus manual chambers in the other
studies).
The best set of meteorological parameters, explaining 53% of the seasonal variation in CH$_4$ fluxes from
upland soils, were SWC, Ts and Rg (Table 2), consistent with H2. We observed a net upland soil CH$_4$
uptake during both the driest and the wettest seasons; CH$_4$ emissions occurred only on a few days
during the wettest season (Fig. 2b). This can likely be explained by the soil characteristics at our site
where upland soils were hypoferralic acrisols, characterised by deep vertical drainage (Epron et al.,
2006). It is likely that these well aerated soils provided the aerobic conditions for methanotrophic CH$_4$
oxidation (Smith et al., 2003). The seasonal variations in net CH$_4$ fluxes were strong (Fig. 3b) with a net
soil CH$_4$ uptake twice as high in the driest season as in the wettest season. This is consistent with the
known dependence of soil CH$_4$ fluxes on topsoil SWC (Fig. 4e; Tables 1, 3): dryer soil conditions favour
soil methanotrophy (Le Mer and Roger, 2001) and wet soils reduce methanotrophic communities and
/ or their activity (Covey and Megonigal, 2019).

**4.2 Ecosystem and upland soil-N$_2$O fluxes**
4.2.1 Seasonal variations in ecosystem N$_2$O fluxes: trends and drivers
The measurements at the Guyaflux wet-tropical-forest site revealed very low N$_2$O fluxes, with an
average net emission of 0.6 ± 0.8 nmolN$_2$O m$^{-2}$ s$^{-1}$ (Fig. 3c; Table 1). Though low, this loss of nitrogen
(N) from the ecosystem is equivalent to approximately one-fifth of the annual atmospheric N
deposition at the site (2.7 kgN$_2$O-N ha$^{-1}$ yr$^{-1}$ here vs. 13 kgN$_2$O-N ha$^{-1}$ yr$^{-1}$ in Van Langenhove et al.,
2020). Compared to other publications on forest ecosystem N$_2$O fluxes from studies based on eddy
covariance techniques, net ecosystem N$_2$O fluxes at our study site were very close to the average fluxes



reported by Stiegler et al. (2023) for a regularly-fertilised Indonesian oil palm plantation (0.7 ± 0.0
$nmolN_2O$ $m^{-2}$ $s^{-1}$ or 0.32 ± 0.003 $gN_2O$-N $m^{-2}$ $yr^{-1}$) but much higher than those reported by Mander et al.
(2021) for a temperate riparian deciduous forest (0.1 $nmolN_2O$ $m^{-2}$ $s^{-1}$ or 87.3 $mgN_2O$-N $m^{-2}$ for the
September 2017 - December 2019 period).
Measured ecosystem-level $N_2O$ fluxes at the Guyaflux site were highly variable, but overall showed
little seasonal variation (means of 0.5 ± 0.8 $nmolN_2O$ $m^{-2}$ $s^{-1}$ and 0.5 ± 0.7 $nmolN_2O$ $m^{-2}$ $s^{-1}$, in the driest
and the wettest seasons, respectively; Table S2), partially supporting H1. These observations fall within
the range of net ecosystem $N_2O$ exchanges measured by eddy covariance reported in an oil palm
plantation in Indonesia, with similar mean $N_2O$ emissions of 0.7 $nmolN_2O$ $m^{-2}$ $s^{-1}$ for both the dry and
wet seasons (Stiegler et al., 2023). Once again, this comparison must be interpreted with extreme
caution even though both studies used the eddy covariance technique as the ecosystems and seasons
concerned were different (a tropical oil palm plantation with strong seasons versus a primary wet
tropical forest).
As with $CH_4$ fluxes, the temporal variability of the $N_2O$ fluxes was very high (Fig. 2c). Contrary to H2,
GAM analyses failed to explain or attribute the observed variations in $N_2O$ fluxes to changes in SWC or
other meteorological drivers ($R^2$ = 0.04, Table 2). Furthermore, daily mean ecosystem $N_2O$ fluxes
switched signs and changed in order of magnitude on short time scales, most likely because these
fluxes are controlled by discontinuous microbial processes (Blagodatsky and Smith, 2012). Yet, we did
find statistically significant, though very weak, relationships between ecosystem $N_2O$ fluxes and SWC,
Ts and Rg, suggesting that the wettest season may provide favourable conditions for soil bacterial $N_2O$
production and plant-mediated $N_2O$ transport, which could contribute to higher net $N_2O$ emissions at
the ecosystem level (Stiegler et al., 2023). It is worth noting that the extent to which trees mediate $N_2O$
emissions is still uncertain; at Guyaflux, within the tower footprint, tree stems in the seasonally flooded
forest emit $N_2O$ while those in the upland forest absorb $N_2O$ (Daniel et al., 2023). Other studies at
Guyaflux and in a lowland tropical rain forest in the Réunion Islands reported that tree stems can
absorb $N_2O$ through as yet unknown mechanisms (Bréchet et al., 2021, 2025; Machacova et al., 2021).



This could indeed counteract the overriding, albeit small, net ecosystem $N_2O$ emissions, suggesting
that the proportion of upland versus seasonal areas should be taken into account.

4.2.2 Seasonal variations in upland soil $N_2O$ fluxes: trends and drivers
In support of H1, $N_2O$ fluxes recorded for the upland soils studied were small, averaging 0.1 ± 0.1
nmol$N_2O$ m$^{-2}$ s$^{-1}$ (Table 1, S2), and slightly higher during the wettest season (0.1 ± 0.1 nmol$N_2O$ m$^{-2}$ s$^{-1}$)
compared to the driest season (0.0 ± 0.1 nmol$N_2O$ m$^{-2}$ s$^{-1}$; Table S2). Our flux values were nine times
smaller than those measured with automated chamber systems in a western Kenyan rainforest (0.9
nmol$N_2O$ m$^{-2}$ s$^{-1}$; Werner et al., 2007), even though the soils in both cases were predominantly $N_2O$
emitters. However, our seasonal $N_2O$ flux observations were within the same order of magnitude as
those in two tropical rainforests where soil $N_2O$ emissions measured with manual chambers were
lower in the dry season than in the wet season (< 0.20 nmol$N_2O$ m$^{-2}$ s$^{-1}$ and 0.34 nmol$N_2O$ m$^{-2}$ s$^{-1}$,
respectively, in Yu et al., 2021; 0.10 nmol$N_2O$ m$^{-2}$ s$^{-1}$ and 0.49 nmol$N_2O$ m$^{-2}$ s$^{-1}$, respectively, in Werner
et al., 2006).
In contrast to H2, none of our GAMs including meteorological drivers (i.e. SWC, Ts, Rg) predicted the
observed seasonal variations in upland soil $N_2O$ fluxes when only the driest and wettest seasons were
taken into account (Table 2; Fig. 5). However, when all the seasons were accounted for, an increase in
SWC appeared to partially explain the increase in soil $N_2O$ fluxes (Figs. 5e, S5d). Although both
nitrification and denitrification can occur simultaneously in various soil microsites (Stevens et al.,
1997), $N_2O$ release often occurs on a daily basis in environments with rapidly shifting $O_2$ availability,
which is the case for soils with changing SWC (Davidson, 1992; Fig. 1). Several studies in subtropical
and tropical forests have reported significant effects of SWC on tropical soil $N_2O$ fluxes (Kiese and
Butterbach-Bahl, 2002; Werner et al., 2006, 2007; Gütlein et al., 2018).



**5  Conclusion**

Our long-term monitoring of ecosystem and soil $CH_4$ and $N_2O$ fluxes over a period of 26 months under contrasting climatic conditions (driest versus wettest seasons) revealed highly variable fluxes that changed direction and amount on short time scales. Although mean daily fluxes were low, $N_2O$ emissions were observed all year long. In contrast, for $CH_4$, either emission or uptake occurred, depending on the season. As expected, the seasons had a statistically significant effect on ecosystem $CH_4$ and $N_2O$ fluxes, with $CH_4$ uptake and higher $N_2O$ emissions during the driest season than during the wettest season. Upland soils exhibited highly variable $CH_4$ and $N_2O$ fluxes, with an increase in $CH_4$ uptake and a decrease in $N_2O$ emissions from the wettest to the driest season. The climatic variables we selected explained only a minor part of the seasonal variations in ecosystem $CH_4$ and $N_2O$ fluxes. At the soil level, none of the climatic variables were significant for seasonal fluxes of $N_2O$ whereas SWC was a strong driver of $CH_4$ fluxes.

Measurements at the ecosystem and soil levels showed divergent fluxes, probably because soil fluxes represent only one compartment in the whole ecosystem. Furthermore, upland soils (52% of the footprint area) are only one type of soil within the large range of soils found inside the Guyaflux tower footprint. In addition, soil chambers provide integrated fluxes for a much smaller area than does the eddy covariance technique. In order to improve the understanding of seasonal variations in ecosystem $CH_4$ and $N_2O$ fluxes, our study shows that it is crucial to characterise the fluxes for all existing ecosystem compartments at the same time and to include all the tree components (leaves, stems, branches) and tree species in the forest habitats, not just those on upland soils. However, our study still provides valuable data that, when combined with mechanistic models, may help identify the missing drivers responsible for the seasonal variations in $CH_4$ and $N_2O$ fluxes in wet tropical forest ecosystems.



**Data availability**

All raw data can be provided by the corresponding authors upon request.

**Supplement link**

**Author contributions**

LMB, MI, CS, DB, IAJ conceived the ideas and designed the methodology; LMB and BB collected the data; LMB and MI performed quality control checks on the data and analysed the data; IAJ and RBJ obtained the funding; LMB led the writing of the manuscript and all authors contributed to the manuscript and gave final approval for submission.

**Competing interests**

The authors declare that they have no conflict of interest.

**Acknowledgements**

We would like to thank Jean-Yves Goret, Nicola Arriga and Elodie Courtois for their technical support.

We thank Vicki Moore for correcting the English of this paper.

**Financial support**

This work was supported by the European Research Council Synergy grant ERC-2013-SyG-610028-IMBALANCE-P and the European Commission through a Marie Skodowska-Curie Individual Fellowship H2020-MSCA-IF-2017-796438 awarded to L. M. Bréchet, the UMR "Ecologie des Forets de Guyane" (EcoFoG) and the Research Fund of the University of Antwerp. This work was also supported by the Gordon and Betty Moore Foundation, Stanford University and the National Research Institute for Agriculture, Food and Environment (INRAE) through the Gordon and Betty Moore Foundation grant



GBMF-11519 for L. M. Bréchet's Postdoctoral Fellowship, and by an Investissement d'Avenir grant from
the Agence Nationale de la Recherche (CEBA: ANR-10-LABX-25-01).

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
