# Peer review of "2 3 Dynamics and environmental drivers of methane and nitrous oxide fluxes at the soil and ecosystem levels in a wet tropical forest 5 6 7 Authors Laëtitia M. Bréchet1, Mercedes Ibáñez1, Robert B. Jackson2, Benoît Burban1, Clément Stahl1"

_EGUsphere, 2025_

## Referee Comment (RC2)

Tropical forest in general are still largely understudied and to understand possible future pathways of these extensive forests, first the current state and drivers need to be understood, which is where studies like this come in play. Lont term data sets of ecosystem and soil fluxes especially of non-co2-greenhouse gases are rare. This study therefore delivers an important and useful contribution to the study of tropical forests. The combination of Eddy covariance (EC) and automated soil chamber measurements for N2O and CH4 in a tropical forest is novel and certainly not trivial to accomplish. The highly variable fluxes with a significant seasonal effect is as expected from n2o and ch4 fluxes in tropical forest.

I agree with referee #1 that the removal of flux values outside the 5th - 95th percentile flux range is a concern. I think it at least needs a strong reasoning and explanation in the text as why the authors chose to do this and which values were removed. Also the sporadic high ch4 emissions and the many n2o uptake events are not discussed, even though they are of interest.

In the attachment I add some comments, a few questions and some suggestions to improve readability and/ or interpretation.

Line 186: automated static non-steady through flow chambers

➔ automated non-steady-state flow-through chambers

Chamber measurements

There are two closure times (2 min and 25 min) and 16 chambers. Could you describe in short how one cycle of all chambers happens? Do the 2 min and 25 min closure time happen shortly after each other for the same chamber, or only specific chambers that close 2 min and other 25 mins? How many measurements per chamber per day? Did you use both closure times in the calculations or did you select one of the two (in caption of Figure S4 is stated that 5 min closure time is used for ch4)? You then link them to the according half hour to match up with the EC measurements. How many measurements per half hour?

Line 212-213: Flux values were selected based on the model that provided the best fit and highest determination coefficient ($R^2$)

How was the best fit determined?

Line 219-221 : As an improvement over Courtois et al. (2019), all CH4 fluxes with R2 < 0.80 were excluded regardless of the measurement length (i.e. 2-min and 25-min). For N2O, all short measurements (i.e. 2-min) with R2 < 0.80 were discarded.

>  If the CO2 measurement during a period is correct, why then still throw out the low R2 measurements for ch4 and n2o? If co2 flux is correct, the low r2 can not be due to analyzer failure or bad closure of the chamber, so then the flux must be correct and might be just small instead of false? Is it not better to put them to 0 instead of remove them out of the data set?

Flux data analysis

Line 248-249 For eddy covariance and chamber data, the 30 min observations were filtered and flux values outside the 5th - 95th percentile flux range were discarded.

>  What do you mean with filtered here and why are the values outside the $5^{th}$-$95^{th}$ percentile thrown out of the data set? Frequently fast increases (high spikes) after rainfall events are seen in N2O soil fluxes (Daelman et al. 2025 https://doi.org/10.5194/bg-22-1529-2025,  Werner et al. 2007 **https://doi.org/10.1029/2006JD007388**), these peaks might be thrown out here.

>  Also sporadic high emissions for CH4 are measured, which also might be partly removed (again Daelman et al. 2025 https://doi.org/10.5194/bg-22-1529-2025 or Barthel et al. 2022 https://doi.org/10.1038/s41467-022-27978-6)

>  Also for the EC data: if the data QA/QC is carefully carried out, it seems that there is no need to remove values outside the $5^{th}$ – $95^{th}$ percentile. However you could opt for a despiking algorithm like the median absolute deviation algorithm (Mauder et al., 2013; Papale et al., 2006).

Line 249 – 257  To calculate daily averages for greenhouse gas fluxes, we first estimated the optimal number of observations per day necessary to obtain representative daily averages. To do this, we selected a data pool with at least 42 observations per day in the eddy covariance dataset. In the soil chamber dataset, we calculated daily means for each of the thirteen chambers and retained only the data when at least five observations per chamber per day were recorded. Subsets of values from 1 to 42 for the eddy covariance data and from 1 to 13 for the soil chamber data were then created for each day based on 100 bootstrap iterations. Representative daily means were found for thresholds of 12 minimum observations per day for eddy covariance and 10 for chamber data.

>  How did you quantify the representativeness?

I think this part is interesting but a bit confusing in the way it is formulated. I suggest some small alterations and I have some questions linked to the way I understand the paragraph. If I just misunderstood the analyses, please ignore the questions that are not relevant.

Alteration: To calculate daily averages for greenhouse gas fluxes, we first estimated the optimal number of observations per day necessary to obtain representative daily averages for EC measurements **and the optimal number of chambers to obtain spatial representative daily averages for chamber measurement**.

→ You differ in amount of chambers in the bootstrap (1 to 13), not the amount of measurements per chamber, which is always at least 5, right? So you want a daily coverage of at least 5 measurements and you are mainly looking for the correct spatial coverage with the bootstrap?

Alteration: **Subsets of each day from the selected data pool were created of sizes 1 to 42 for the eddy covariance data and subsets of daily averages of size1 to 13 for the soil chamber data were then created for each day based on 100 bootstrap iterations.**

Could you include anything about the diel cycle of the EC measurements. Many more night time measurements will be missing compared to day time measurements, so when taking only 12 half hourly measurements, a large percentage of this will be daytime measurements. As only these 12 half hourly measurements are enough to represent a daily average, I suspect that the diel cycle is not that pronounced.

Could you also include something about the spatial variability of the chambers? As you need 10 out of 13 chambers to achieve representative values, does this mean that there is a high spatial variation between the chambers? Or does this mean that you need a lot of chambers because they are al measured in different half hours and you need to cover many half hours and thus the temporal coverage rather than the spatial coverage is important?

Line 331 – 335 In contrast to the ecosystem-level fluxes, soil $CH_4$ fluxes in some of the upland forest were mainly negative, indicating net soil $CH_4$ uptake throughout the year (Fig. 2b), even under varying environmental conditions. Soil $CH_4$ uptake did decreased significantly in the wettest season compared to the driest season, although the fluxes remained negative overall (i.e. $CH_4$ uptake, Table 1; Fig. 3b).

There are however many high emissions periods for several chambers at different times for ch4. These are not discussed in the article.

Line 342- 345 In contrast to the ecosystem-level fluxes, soil N2O fluxes in upland areas not only had a more pronounced seasonal pattern, the upland soils also emitted more N2O during the wettest season than during the driest season, when the average flux was near-zero N2O (Table 1; Fig. 3d).

There are however many uptake events for N2O, these are also not discussed in the article even though soil uptake of N2O is not that frequently measured in other tropical forest.

Line 559 - Measurements at the ecosystem and soil levels showed divergent fluxes, probably because soil fluxes represent only one compartment in the whole ecosystem. Furthermore, upland soils (52% of the footprint area) are only one type of soil within the large range of soils found inside the Guyaflux tower footprint. In addition, soil chambers provide integrated fluxes for a much smaller area than does the eddy covariance technique.

Just a very small suggestion, not really a must, might be something to think about: This is, I think, the only section where the two kind of measurements (soil chambers and EC) are mentioned together. As stated, the soil chambers are only one compartment of the ecosystem and have a small area so I realize a real comparison or upscaling can not be made, but is there some way both methods could be linked more? I am thinking of a small overview or repetition of what is comparable, what is not and which elements may cause the divergence. Or in someway how these soil chambers fit in the results of the EC. I think there are many elements in the text that point to connection or divergence (like uptake of stem/canopy, different soils, SWC, ....) but somehow the connection between the measurements is lost for me because these elements are spread out in the text. Anyway, not easy to do, just a thought.

---

## Author Response (AR1)

egusphere-2025-3501

**Dynamics and environmental drivers of methane and nitrous oxide fluxes at the soil and ecosystem levels in a wet tropical forest**

**Author(s): Laëtitia Bréchet et al.**

**BIOGEOSCIENCE – Comment on egusphere-2025-3501 – 25/08/2025**
**Response to comments by Anonymous Referee #1**
The author's comments and responses are written in blue, and the comments of Anonymous Referee #1 are in black.

Overall, this paper is a useful addition to the literature. Better characterizing greenhouse gas fluxes from tropical soils, and identifying drivers, is a timely and important research question. The combination of automated chambers and eddy-flux, especially for $N_2O$, is very novel for tropical forests. Overall, given the well-known spatial and temporal heterogeneity of $N_2O$ fluxes even in much more homogenous ecosystems, I am not surprised that fluxes were not well explained by simple environmental variables, even with the data density of this paper. This is a dataset that will certainly be of interest to many people. Several of the methodological lessons learned (such as lack of storage of $N_2O$ and $CH_4$ under the canopy, revealed by vertical profile measurements) are also likely to be of use to other researchers.

**Response:** We thank Referee #1 very much for his / her positive feedback and constructive suggestions.

I have one significant concern that I think merits serious consideration- the choice to exclude high fluxes from analysis (and indeed not to present them at all, making it very difficult to judge how important they might be). Specifically, the authors did not present (or include in analysis) any fluxes outside the 5th-95th percentile (per line 248), even after rigorous data cleaning steps that should have weeded out any anonymously high fluxes that were methodological artifacts. I realize that it is very difficult to scale rare, high fluxes without very good estimates of their probability. However, I do not thing that dismissing them entirely makes any sense, and no specific rationale or citation was given for the choice. Rare, very high $N_2O$ fluxes are not at all uncommon in tropical forests in my personal experience, but rarely do we have the data density (as we do here!) to judge their potential importance. These excluded fluxes- depending on their magnitude, could be potentially important for net ecosystem emissions, especially because they're probably quite skewed- extreme production events could be somewhat common but extreme consumption events likely are not.

To summarize- I am not necessarily suggesting that all data needs to incorporated into scaling, but I would strongly suggest 1) presenting the relative magnitude of the excluded fluxes compared to the data that was included. Were they common and somewhat high? Or more rare and extremely high? 2) At least conducting some sort of sensitivity of means, medians etc to the inclusion or exclusion of these omitted values (including perhaps making a range of assumptions about their probability, in the case of very high outliers). I also would hazard, and add caveats, against comparisons with any other rate fluxes from tropical sites that may indeed have included hot spots and hot moments in their scaling efforts. Overall it seems counterintuitive to highlight the heterogeneity of soil GHG fluxes and then ignore a potential large fraction of the variation.

**Response:** This is a very relevant concern, particularly given the highly variable nature of the $CH_4$ and $N_2O$ fluxes. For clarity, we now present the average daily fluxes based on the raw data (fluxes computed by

the EDDYPRO and SOILFLUXPRO programs), rather than the 5th-95th percentile range. Overall, it has no impact at the ecosystem level. Figure 1 shows that considering the raw data has little effect on the calculated daily fluxes of CH4 at the ecosystem and soil levels. Only a few episodic production events (two in July 2016 and June 2018) and no remarkable consumption events are observed at the ecosystem level. Meanwhile, high N2O fluxes are much more frequent at the soil level throughout the study period, although this is not so visible at the ecosystem level.

(A) Ecosystem level

[Figure]

(B) Soil level

[Figure]

**Figure 1**. Seasonal courses of average daily (A) ecosystem and (B) soil fluxes for CH4 (left-hand panel) and N2O (right-hand panel), from 17 May, 2016 to 2 August, 2018 in the Guyaflux tropical forest, French Guiana. Red dots represent the average daily fluxes calculated with the raw data and black dots those calculated with the 5th-95th percentile data.

To summarise, regarding your two suggestions:

1) See Figure 1 for a comparison of flux databases without and with high fluxes (black and red dots, respectively). The high fluxes are particularly common, but not so high, for soil $N_2O$ fluxes, while they are comparably very rare and extremely high for ecosystem $CH_4$ fluxes. These observations will be included in the Discussion section.
2) To complement this, see Table 1 for descriptive statistics (means, standard deviations, and medians) showing the effect of capping thresholds on our main analytical results in the wettest and driest seasons in particular. Major changes, highlighted in light green. At the ecosystem level, the two peaks in $CH_4$ occurred during the wettest season, with no effect on the mean and median values. However, during the driest season, more consumption events tend to lower the mean and median values. Regarding $N_2O$ fluxes, as mentioned above, although there are no extreme peaks, high fluxes tend to increase $N_2O$ emissions at soil level, but have no noticeable effect at ecosystem level.

**Table 1**. A), B) Mean, standard deviation (SD) and median ecosystem and upland soil $CH_4$ and $N_2O$ fluxes for the wettest and driest seasons in the Guyaflux tropical forest, French Guiana.

A) BEFORE: Preprint databases

| Fluxes | Wettest | | | Driest | | |
|---|---|---|---|---|---|---|
| | Mean | SD | Median | Mean | SD | Median |
| **Ecosystem flux (nmol$_{CH4/N2O}$ m$^{-2}$ s$^{-1}$)** | | | | | | |
| CH₄ | 2.9 | 3.9 | 2.8 | -0.8 | 3.8 | -0.6 |
| N₂O | 0.5 | 0.7 | 0.4 | 0.5 | 0.8 | 0.6 |
| **Upland soil flux (nmol$_{CH4/N2O}$ m$^{-2}$ s$^{-1}$)** | | | | | | |
| CH₄ | -0.8 | 0.5 | -0.8 | -1.8 | 0.5 | -1.8 |
| N₂O | 0.1 | 0.1 | 0.1 | 0.0 | 0.1 | 0.0 |

B) AFTER: Revised databases

| Fluxes | Wettest | | | Driest | | |
|---|---|---|---|---|---|---|
| | Mean | SD | Median | Mean | SD | Median |
| **Ecosystem flux (nmol$_{CH4/N2O}$ m$^{-2}$ s$^{-1}$)** | | | | | | |
| CH₄ | 4.9 | 11.2 | 3.5 | -1.6 | 6.4 | -1.7 |
| N₂O | 0.6 | 1.0 | 0.5 | 0.5 | 1.3 | 0.7 |
| **Upland soil flux (nmol$_{CH4/N2O}$ m$^{-2}$ s$^{-1}$)** | | | | | | |
| CH₄ | -0.4 | 0.9 | -0.6 | -1.4 | 1.2 | -1.8 |
| N₂O | 0.3 | 0.3 | 0.3 | 0.2 | 0.3 | 0.2 |

Finally, Figure 2 shows all the daily mean fluxes, calculated from the raw data. This is the same as Figure 1, but with the colours indicating the different seasons and showing the conservative 1st and 99th percentiles.

[Figure]

**Figure 2**. Seasonal courses of the raw average daily a) ecosystem and b) soil fluxes for $CH_4$ (top panels) and $N_2O$ (bottom panels) for the full datasets from 17 May, 2016 to 2 August, 2018 in the Guyaflux tropical forest, French Guiana. The 1st-99th percentile ranges of the flux values are represented by the horizontal dashed lines. Colours illustrate the wet, intermediate, and dry seasons, and for two contrasted seasons, defined as the wettest (dark blue dots) and the driest (red dots).

We will revise the text to include other rate fluxes from tropical sites that have incorporated hotspots and hot moments into their scaling efforts, e.g. Dealman et al., (2025).

Daelman, R., Bauters, M., Barthel, M., Bulonza, E., Lefevre, L., Mbifo, J., ... & Boeckx, P., 2025. Spatiotemporal variability of $CO_2$, $N_2O$ and $CH_4$ fluxes from a semi-deciduous tropical forest soil in the Congo Basin. Biogeosciences. 22, 1529–1542.

Finally, data should be posted in an accessible database online in keeping with the specifications of this journal (rather than 'on request')

**Response:** We fully agree. Data status will be updated and we will provide in the final version of this manuscript a link (e.g. Zenodo) for online access to all data included here

The author's comments and responses are written in blue, and the comments of Anonymous Referee #2 are in black.

Tropical forest in general are still largely understudied and to understand possible future pathways of these extensive forests, first the current state and drivers need to be understood, which is where studies like this come in play. Long-term data sets of ecosystem and soil fluxes especially of non-co2-greenhouse gases are rare. This study therefore delivers an important and useful contribution to the study of tropical forests. The combination of Eddy covariance (EC) and automated soil chamber measurements for $N_2O$ and $CH_4$ in a tropical forest is novel and certainly not trivial to accomplish. The highly variable fluxes with a significant seasonal effect is as expected from $N_2O$ and $CH_4$ fluxes in tropical forest.

**Response**: We thank the referee very much for this positive feedback.

I agree with referee #1 that the removal of flux values outside the $5^{th}$ - $95^{th}$ percentile flux range is a concern. I think it at least needs a strong reasoning and explanation in the text as why the authors chose to do this and which values were removed. Also the sporadic high $CH_4$ emissions and the many $N_2O$ uptake events are not discussed, even though they are of interest.

**Response**: This is a pertinent concern, especially given the highly variable nature of the $CH_4$ and $N_2O$ fluxes. For clarity, the average daily fluxes are now calculated based on the raw data (i.e. the data computed by the EDDYPRO and SOILFLUXPRO programs) rather than data within the 5th-95th percentile range only. Overall, it has no impact at the ecosystem level. Figure 1 in Referee #1's response shows that considering the raw data has little effect on the calculated daily fluxes of $CH_4$ at the ecosystem and soil levels. Only a few episodic production events (two in July 2016 and June 2018) and no remarkable consumption events are observed at the ecosystem level. Meanwhile, high (though not extreme) $N_2O$ fluxes are much more frequent at the soil level throughout the study period, although they have no statistical impact at the ecosystem level (see Table 2 below). These observations will be included in the Discussion section, although the overall message of the manuscript remains unchanged. The revised graphs below (corresponding to Figures 2 and 3 in the preprint) present the average daily flux datasets based on the raw values.

We have addressed all of the more specific comments in the attachment below.

[Figure]

**Figure 1**. Seasonal courses of average daily ecosystem fluxes (crosses on the left) and upland soil fluxes (solid dots on the right), calculated from raw data, for the wet, intermediate and dry seasons, and for two contrasted seasons defined as the wettest (dark blue dots) and the driest (red dots) for 24-hour $CH_4$ fluxes (a, b) and $N_2O$ fluxes from 17 May, 2016 to 2 August, 2018 in the Guyaflux tropical forest, French Guiana. The dashed line represents the zero line and the solid line represents the overall average.

[Figure]

**Figure 2**. Boxplots and associated density plots of average daily ecosystem fluxes (dashed lines on the left) and upland soil fluxes (solid lines on the right), calculated from raw data, of 24-hour $CH_4$ fluxes (a, b) and $N_2O$ fluxes (c, d) for the wettest (blue) and driest (red) seasons, from 17 May, 2016 to 2 August, 2018 in the Guyaflux tropical forest, French Guiana.

In the atta        chment I add some comments, a few questions and some suggestions to improve readability and / or interpretation.
* * *
Attachment file

Line 186: automated static non-steady through flow chambers
- automated non-steady-state flow-through chambers

**Response:** We revised this statement as suggested.

Chamber measurements
There are two closure times (2 min and 25 min) and 16 chambers. Could you describe in short how one cycle of all chambers happens? Do the 2 min and 25 min closure time happen shortly after each other for the same chamber, or only specific chambers that close 2 min and other 25 mins? How many measurements per chamber per day? Did you use both closure times in the calculations or did you select one of the two (in caption of Figure S4 is stated that 5 min closure time is used for $CH_4$)? You then link them to the according half hour to match up with the EC measurements. How many measurements per half hour?

**Response:** Thank you for pointing out that this was not entirely clear. The following details of the soil flux system will be added in the main text of the manuscript:

Line 204-207: "In addition, to maximise the percentage of fluxes that can be detected for $N_2O$ and $CH_4$ without affecting spatial coverage and temporal resolution, we initially developed a program combining two different closure times. Briefly, each week, four of the initial sixteen chambers were manually set to remain closed for 25 minutes, while the other chambers were set to remain closed for a much shorter period of 2 minutes. This program was rotated across the chambers. The 25-min closure time was a compromise between reliably estimating low $N_2O$ fluxes and ensuring a sufficient number of flux measurements per chamber and per day. The measurement cycle duration was 2.5 hours, providing approximately nine measurements per chamber per day for all gases. Each chamber was therefore measured with the longer closure time for one 7-consecutive-day period per month (4 weeks). More detail of equipment can be found in previous publications (Courtois et al., 2019; Bréchet et al., 2021)."

Thank you also for checking the documents in detail; the captions of Figures S4 and S5 will be corrected as follows: "Fluxes were estimated with a 2-min and 25-min closure time."

Line 212-213: Flux values were selected based on the model that provided the best fit and highest determination coefficient ($R^2$)
- How was the best fit determined?

**Response:** Thank you for highlighting that this was unclear. We will remove this sentence, as explanations are provided in the paragraph (Line 215-225).

Line 219-221 : As an improvement over Courtois et al. (2019), all $CH_4$ fluxes with $R^2 < 0.80$ were excluded regardless of the measurement length (i.e. 2-min and 25-min). For $N_2O$, all short measurements (i.e. 2-min) with $R^2 < 0.80$ were discarded.

- If the $CO_2$ measurement during a period is correct, why then still throw out the low $R^2$ measurements for $CH_4$ and $N_2O$? If $CO_2$ flux is correct, the low $R^2$ can not be due to analyzer failure or bad closure of the chamber, so then the flux must be correct and might be just small instead of false? Is it not better to put them to 0 instead of remove them out of the data set?

**Response:** We believe it raises the overall difficulty of treating the highly variable, low background atmospheric concentrations of $CH_4$ and $N_2O$, which we know originate from different processes to $CO_2$. Beyond that, we realised that the sentence was incorrect, since no criteria from the $R^2$ had been applied to the $N_2O$ flux values. However, as a low $R^2$ value corresponding to a poor goodness-of-fit measure of the regression model does not necessarily indicate a low slope and therefore a near-zero flux, we do not believe it would be correct to set all the fluxes with low $R^2$ at 0. Finally, the paragraph about the minimum detection limits mentioned in the article was also not entirely clear. It will be revised as follows:

Line 215-225: "After calculating the fluxes and implementing our standard soil greenhouse gas QC procedure (Courtois et al., 2019; Bréchet et al., 2021), all $CO_2$ fluxes with an insufficiently high $R^2$ (< 0.90), an initial concentration greater than 900 ppm, or a value outside the range of variation from 0.10 to 30 $\mu mol\ m^{-2}\ s^{-1}$ were discarded for all three gases, based on the assumption that poor-quality $CO_2$ implied poor-quality values for $CH_4$ and $N_2O$. Based on previous work (Courtois et al., 2019), all $CH_4$ fluxes with $R^2 < 0.80$ were excluded (17% of the total 89380 values). These fluxes were not set to zero, as a low $R^2$ value does not provide information on the flux value itself. This criterion was inapplicable to $N_2O$, however, due to the low overall $R^2$ values, meaning all fluxes were retained for the study. Additionally, we computed the minimum detectable fluxes (MDF) suitable for high-resolution, in situ greenhouse gas measurements, using the metric proposed by Nickerson (2016): 0.040 nmol $m^{-2}\ s^{-1}$ and 0.002 nmol $m^{-2}\ s^{-1}$ for 2 min and 25 min, respectively, for $CH_4$; and 0.100 nmol $m^{-2}\ s^{-1}$ and 0.002 nmol $m^{-2}\ s^{-1}$ for 2 min and 25 min, respectively, for $N_2O$. This implies that null fluxes were included in the analysis if the flux values for each gas and measurement length fell within the range defined by the corresponding absolute MDF value."

By considering low fluxes as null values rather than "NA" (not a number), based on the MDF thresholds, we additionally retained:
- 24 values of soil $CH_4$ fluxes,
- 11994 values of soil $N_2O$ fluxes.

As shown in Table 1, there is no substantial effect on the descriptive statistics.

**Table 1**. Descriptive statistics for soil $CH_4$ and $N_2O$ fluxes when flux values within the absolute MDF value ranges were excluded (left-hand panel), and when they were included in the analysis as null fluxes (right-hand panel).

| Summary table with low fluxes = "NA" | | Summary table with low fluxes = 0 | |
|---|---|---|---|
| **FluxCH4** | | **FluxCH4** | |
| Mean (SD) | 0.0196 (0.370) | Mean (SD) | 0.0196 (0.370) |
| Median [Min, Max] | -0.00124 [-0.0349, 17.4] | Median [Min, Max] | -0.00124 [-0.0349, 17.4] |
| Missing | 27416 (29.6%) | Missing | 27392 (29.6%) |
| **FluxN2O** | | **FluxN2O** | |
| Mean (SD) | 0.000681 (0.0916) | Mean (SD) | 0.000567 (0.0836) |
| Median [Min, Max] | 0.00000395 [-0.0783, 22.5] | Median [Min, Max] | 0 [-0.0783, 22.5] |
| Missing | 32036 (34.6%) | Missing | 20042 (21.7%) |

Flux data analysis

Line 248-249 For eddy covariance and chamber data, the 30 min observations were filtered and flux values outside the 5th-95th percentile flux range were discarded.

- What do you mean with filtered here and why are the values outside the 5[th]-95[th] percentile thrown out of the data set? Frequently fast increases (high spikes) after rainfall events are seen in $N_2O$ soil fluxes (Daelman et al. 2025, https://doi.org/10.5194/bg-22-1529-2025; Werner et al. 2007, https://doi.org/10.1029/2006JD007388), these peaks might be thrown out here.

  Also sporadic high emissions for $CH_4$ are measured, which also might be partly removed (again Daelman et al. 2025, https://doi.org/10.5194/bg-22-1529-2025 or Barthel et al. 2022, https://doi.org/10.1038/s41467-022-27978-6).

  Also for the EC data: if the data QA/QC is carefully carried out, it seems that there is no need to remove values outside the 5th-95th percentile. However you could opt for a despiking algorithm like the median absolute deviation algorithm (Mauder et al. 2013; Papale et al. 2006).

**Response**: As mentioned above, for clarity, we now present the average daily fluxes based on the raw data (fluxes computed by the EDDYPRO and SOILFLUXPRO programs), rather than the 5[th]-95[th] percentile range.

To summarise, Figure 1 (in Referee #1's response), comparing the flux databases without and with high fluxes (black and red dots, respectively), shows that high fluxes are particularly common, but not so high for soil $N_2O$ fluxes, while they are comparably very rare and extremely high for ecosystem $CH_4$ fluxes. These observations will be included in the Discussion section. Additionally, Table 2 illustrates the effect of capping thresholds on our main analytical results during the wettest and driest seasons. Major changes are highlighted in light green. At the ecosystem level, the two extreme peaks in $CH_4$ occurred during the wettest season, with no effect on the mean and median values. However, during the driest season, the mean and median values tend to be lower due to more consumption events. Regarding $N_2O$ fluxes, as mentioned above, although there are no extreme peaks, retained high fluxes tend to increase $N_2O$ emissions at soil level, but have no noticeable effect at ecosystem level.

**Table 2**. A), B) Mean, standard deviation (SD) and median ecosystem and upland soil $CH_4$ and $N_2O$ fluxes for the wettest and driest seasons in the Guyaflux tropical forest, French Guiana.

A) BEFORE: Preprint databases

| Fluxes | Wettest | | | Driest | | |
|---|---|---|---|---|---|---|
| | Mean | SD | Median | Mean | SD | Median |
| **Ecosystem flux (nmol$_{CH4/N2O}$ m$^{-2}$ s$^{-1}$)** | | | | | | |
| $CH_4$ | 2.9 | 3.9 | 2.8 | -0.8 | 3.8 | -0.6 |
| $N_2O$ | 0.5 | 0.7 | 0.4 | 0.5 | 0.8 | 0.6 |
| **Upland soil flux (nmol$_{CH4/N2O}$ m$^{-2}$ s$^{-1}$)** | | | | | | |
| $CH_4$ | -0.8 | 0.5 | -0.8 | -1.8 | 0.5 | -1.8 |
| $N_2O$ | 0.1 | 0.1 | 0.1 | 0.0 | 0.1 | 0.0 |

B) AFTER: Revised databases

| Fluxes | Wettest | | | Driest | | |
|---|---|---|---|---|---|---|
| | Mean | SD | Median | Mean | SD | Median |
| **Ecosystem flux (nmol$_{CH4/N2O}$ m$^{-2}$ s$^{-1}$)** | | | | | | |
| $CH_4$ | 3.3 | 6.2 | 3.3 | -1.7 | 6.4 | -1.8 |
| $N_2O$ | 0.6 | 1.0 | 0.5 | 0.6 | 1.2 | 0.7 |
| **Upland soil flux (nmol$_{CH4/N2O}$ m$^{-2}$ s$^{-1}$)** | | | | | | |
| $CH_4$ | -0.5 | 0.8 | -0.7 | -1.4 | 1.2 | -1.8 |
| $N_2O$ | 0.3 | 0.3 | 0.3 | 0.2 | 0.3 | 0.2 |

Line 249-257 To calculate daily averages for greenhouse gas fluxes, we first estimated the optimal number of observations per day necessary to obtain representative daily averages. To do this, we selected a data pool with at least 42 observations per day in the eddy covariance dataset. In the soil chamber dataset, we calculated daily means for each of the thirteen chambers and retained only the data when at least five observations per chamber per day were recorded. Subsets of values from 1 to 42 for the eddy covariance data and from 1 to 13 for the soil chamber data were then created for each day based on 100 bootstrap iterations. Representative daily means were found for thresholds of 12 minimum observations per day for eddy covariance and 10 for chamber data.

- How did you quantify the representativeness?

**Response:** We quantify the representativeness by estimating the minimum sample size of eddy covariance and chamber flux data at which the average daily $CH_4$ and $N_2O$ fluxes stabilise (see Figures 3-5). To achieve this, we calculated the average daily values based on 100 random iterations of flux data subsets (raw data from 2.3 years), ranging from one to 48 samples for the eddy covariance system (30-minute flux values), and from one to 13 samples for the chamber system (with chambers having a minimum of five flux values). The results showed that at least twelve flux values were needed to capture sub-daily variations in eddy covariance (Figures 3), and at least ten chambers were needed to account for sub-daily, spatio-temporal variations in the chambers (Figures 4, 5) at our study site during both the wettest and driest periods.

Note that in their study on the characterisation of errors in methane fluxes and budgets derived from a long-term comparison of open- and closed-path eddy covariance systems, Deventer et al. (2019) based valid 30-min $CH_4$ fluxes on an extrapolated daily value with at least eight observations per day.

These graphs can be included in the Supplement if necessary.

[Figure]

**Figure 3**. Average daily net ecosystem $CH_4$ (left-hand panel) and $N_2O$ (right-hand panel) fluxes, calculated based on 100 random iterations on subsets of 30-min flux data ranging from one to 48 samples. In the box plots, solid bold lines represent medians, box boundaries mark the 25th and 75th percentiles, and whiskers show the 10th and 90th percentiles. Dots mark outliers. The horizontal red line illustrates the overall average, and the vertical dotted blue line the estimated minimum sample size (i.e. 12).

[Figure]

[Figure]

**Figure 4**. Average daily net soil $CH_4$ fluxes, calculated based on 100 random iterations on subsets of flux data ranging from one to 13 samples. In the box plots, solid bold lines represent medians, box boundaries mark the 25th and 75th percentiles, and whiskers show the 10th and 90th percentiles. Dots mark outliers. The horizontal black line illustrates the overall average, and the vertical dotted blue line the estimated minimum sample size (i.e. 10).

[Figure]

**Soil chambers**
**N₂O fluxes**

**Figure 5**. Average daily net soil N₂O fluxes, calculated based on 100 random iterations on subsets of flux data ranging from one to 13 samples. In the box plots, solid bold lines represent medians, box boundaries mark the 25th and 75th percentiles, and whiskers show the 10th and 90th percentiles. Dots mark outliers. The horizontal black line illustrates the overall average, and the vertical dotted blue line the estimated minimum sample size (i.e. 10).

Deventer, M.J., Griffis, T.J., Roman, D.T., Kolka, R.K., Wood, J.D., Erickson, M., ... & Millet, D.B., 2019. Error characterization of methane fluxes and budgets derived from a long-term comparison of open-and closed-path eddy covariance systems. Agric. For. Meteorol. 278, 107638.

I think this part is interesting but a bit confusing in the way it is formulated. I suggest some small alterations and I have some questions linked to the way I understand the paragraph. If I just misunderstood the analyses, please ignore the questions that are not relevant.

Alteration: To calculate daily averages for greenhouse gas fluxes, we first estimated the optimal number of observations per day necessary to obtain representative daily averages for EC measurements **and the optimal number of chambers to obtain spatial representative daily averages for chamber measurement.**

- You differ in amount of chambers in the bootstrap (1 to 13), not the amount of measurements per chamber, which is always at least 5, right? So you want a daily coverage of at least 5 measurements and you are mainly looking for the correct spatial coverage with the bootstrap?
  Alteration: **Subsets of each day from the selected data pool were created of sizes 1 to 42 for the eddy covariance data and subsets of daily averages of size1 to 13 for the soil chamber data were then created for each day based on 100 bootstrap iterations.**

**Response**: We thank Referee #2 for these suggestions, which strongly improved this part. The paragraph will be revised as follows:

Line 249-257: "To calculate daily averages for greenhouse gas fluxes, we first estimated the optimal number of observations per day necessary to obtain representative daily averages. To do this, we selected a data pool with at least 42 observations per day in the eddy covariance dataset. In the soil chamber dataset, we firstly calculated daily means based on a minimum of five observations for each of the thirteen chambers. Then, for each day, we created subsets of values from 1 to 42 for the eddy covariance data, and from 1 to 13 for the soil chamber data, based on 100 bootstrap iterations. Representative daily means were found for minimum thresholds of 12 observations per day for the eddy covariance and 10 chambers per day for the soil flux system."

Could you include anything about the diel cycle of the EC measurements. Many more night time measurements will be missing compared to day time measurements, so when taking only 12 half hourly measurements, a large percentage of this will be daytime measurements. As only these 12 half hourly measurements are enough to represent a daily average, I suspect that the diel cycle is not that pronounced.

**Response**: This is an excellent point. Figure 6 shows that our eddy flux measurements more accurately reflect daytime variations than night-time variations (i.e. fewer observations at night compared to day). The systematic bias error associated with the eddy covariance technique (i.e. u* filtering) perceived underestimation of nocturnal ecosystem efflux during low wind conditions has a known effect on ecosystem $CO_2$ fluxes, but this is less clear for $CH_4$ and $N_2O$, since the processes are different. So, we are aware that this may introduce a bias when calculating the average daily fluxes. However, we currently do not know how to solve this issue, since we do not know how to fill the gaps in the 30-min $CH_4$ and $N_2O$ flux values to calculate daily means. Instead, we applied a minimum threshold of 12 observations to be representative of sub-daily variations at our study site (see our response above and Figures 3-5).

[Figure]

**Figure 6**. Graph showing the number of eddy flow observations per hour.

Could you also include something about the spatial variability of the chambers? As you need 10 out of 13 chambers to achieve representative values, does this mean that there is a high spatial variation between the chambers? Or does this mean that you need a lot of chambers because they are al measured in different half hours and you need to cover many half hours and thus the temporal coverage rather than the spatial coverage is important?

**Response**: In fact, ten out of thirteen soil chambers enable the spatial variability of $CH_4$ and $N_2O$ fluxes to be taken into account while still providing good temporal representativeness of these fluxes throughout the day. Figures S4 and S5 in the Supplement present the average daily soil flux values for each of the thirteen chambers for $CH_4$ and $N_2O$, respectively. These figures reveal significant spatial variation between the chambers, which can be mitigated by keeping values from at least ten chambers. More details on the chamber measurements will be provided in Section 2.4 (see above), along with information on the associated spatial variation at soil level. Figures S4 and S5 will be revised in line with our response to Referee #1 (see Figure 2 for the full dataset, which includes black and red dots).

Line 331-335 In contrast to the ecosystem-level fluxes, soil $CH_4$ fluxes in some of the upland forest were mainly negative, indicating net soil $CH_4$ uptake throughout the year (Fig. 2b), even under varying environmental conditions. Soil $CH_4$ uptake did decreased significantly in the wettest season compared to the driest season, although the fluxes remained negative overall (i.e. $CH_4$ uptake, Table 1; Fig. 3b).
- There are however many high emissions periods for several chambers at different times for $CH_4$. These are not discussed in the article.

**Response:** We agree and will therefore provide more information in the Discussion. Indeed, we can only speculate about these differences. Microtopographic and edaphic heterogeneity may have caused some chamber locations to remain aerobic in surface horizons, even during the wettest season. This may explain their year-round $CH_4$-uptake, albeit reduced during the wettest season. Other locations, in contrast, may have become anaerobic during the wet season, disabling methanotrophs to oxidize the produced $CH_4$ and eliciting a switch to net $CH_4$ efflux.

Line 342-345 In contrast to the ecosystem-level fluxes, soil $N_2O$ fluxes in upland areas not only had a more pronounced seasonal pattern, the upland soils also emitted more $N_2O$ during the wettest season than during the driest season, when the average flux was near-zero $N_2O$ (Table 1; Fig. 3d).
- There are however many uptake events for $N_2O$, these are also not discussed in the article even though soil uptake of $N_2O$ is not that frequently measured in other tropical forest.

**Response:** We agree and will include more information in the Discussion section. It is indeed true that net $N_2O$ uptake is rarely reported for tropical forests. Previous observations at our study site showed not only soil $N_2O$ uptake, but also that these fluxes, although low, were mainly recorded over short measurement periods (2 minutes instead of 25 minutes; Courtois et al., 2019). It is possible that the very low flux detection limit of our instruments (G2308; Picarro) explains why we frequently measured net uptake fluxes. Since $N_2O$ uptake rates are typically small, setups with a high MDF (minimum detectable flux), which are usually employed in remote areas, such as tropical rainforests, are often unable to detect these small uptake fluxes.
Regarding the seasonal pattern, we suspect that, during the wet season, when conditions are ideal for both litter mineralisation and heterotrophic $N_2$ fixation (see Van Langenhove et al., 2020), as well as for denitrification, $N_2O$ production typically exceeds $N_2O$ consumption. During the dry season, however, the drier conditions in the topsoil are more favourable for $N_2O$ uptake, enabling microbial communities to consume all the $N_2O$ produced (on average).

Courtois, E.A., Stahl, C., Burban, B., Van den Berge, J., Berveiller, D., Bréchet, L., ... & Janssens, I.A., 2019. Automatic high-frequency measurements of full soil greenhouse gas fluxes in a tropical forest. Biogeosciences. 16, 785-796.
Van Langenhove, L., Depaepe, T., Vicca, S., Van den Berge, J., Stahl, C., Courtois, E., ... & Janssens, I. A., 2020. Regulation of nitrogen fixation from free-living organisms in soil and leaf litter of two tropical forests of the Guiana shield. Plant Soil. 450, 93-110.

Line 559-Measurements at the ecosystem and soil levels showed divergent fluxes, probably because soil fluxes represent only one compartment in the whole ecosystem. Furthermore, upland soils (52% of the footprint area) are only one type of soil within the large range of soils found inside the Guyaflux tower footprint. In addition, soil chambers provide integrated fluxes for a much smaller area than does the eddy covariance technique.

- Just a very small suggestion, not really a must, might be something to think about: This is, I think, the only section where the two kind of measurements (soil chambers and EC) are mentioned together. As stated, the soil chambers are only one compartment of the ecosystem and have a small area so I realize a real comparison or upscaling can not be made, but is there some way both methods could be linked more? I am thinking of a small overview or repetition of what is comparable, what is not and which elements may cause the divergence. Or in someway how these soil chambers fit in the results of the EC. I think there are many elements in the text that point to connection or divergence (like uptake of stem/canopy, different soils, SWC,….) but somehow the connection between the measurements is lost for me because these elements are spread out in the text. Anyway, not easy to do, just a thought.

**Response:** This is indeed an issue in studies that explore fluxes at different spatial scales. Ideally, we would conduct a more in-depth analysis by making direct comparisons of the data from the two levels and providing further interpretation of the results. However, as mentioned in the manuscript, this requires additional data from complementary studies, because no matter how small the tower's footprint, it always includes fluxes derived from the seasonally flooded area.
Here are some additional explanations. The Guyaflux experimental unit covers over 400 ha of undisturbed forest, and the tower measures fluxes from over 1 km of primary forest in the direction of the prevailing winds. Bonal et al. (2008) compared the effects of the dry and wet seasons on net ecosystem exchange in the Guyaflux forest, and showed that wind direction and speed (the two main determinants of footprint size and location) did not yield statistically different fluxes. Regardless of the size of the footprint, the creek and flooded areas always make a substantial contribution. Consequently, while it would be insightful to distinguish between the wet and dry parts of the forest or between footprints that include or exclude flooded areas, this is not feasible.

We will revise the conclusion to encompass the fact that the eddy covariance technique used at the tower provides flux data resulting from integrated processes across various forest habitats, soil types, tree species, hydrological conditions, and topographical positions. In addition, the tower's footprint covers a much larger area than the deployed soil flux chamber system. The combination of these different conditions may explain the highly heterogeneous soil $CH_4$ and $N_2O$ fluxes at landscape scales. However, this makes comparison between the two measurement techniques rather hazardous.